# Mammalian histones facilitate antimicrobial synergy by disrupting the bacterial proton gradient and chromosome organization

Tory Doolin[1], Henry M. Amir [2], Leora Duong[3], Rachel Rosenzweig [4], Lauren A. Urban[5], Marta Bosch[6,7], Albert Pol [6,7,8], Steven P. Gross[1,2✉] & Albert Siryaporn [2,3✉]

First proposed as antimicrobial agents, histones were later recognized for their role in condensing chromosomes. Histone antimicrobial activity has been reported in innate immune responses. However, how histones kill bacteria has remained elusive. The co-localization of histones with antimicrobial peptides (AMPs) in immune cells suggests that histones may be part of a larger antimicrobial mechanism in vivo. Here we report that histone H2A enters *E. coli* and *S. aureus* through membrane pores formed by the AMPs LL-37 and magainin-2. H2A enhances AMP-induced pores, depolarizes the bacterial membrane potential, and impairs membrane recovery. Inside the cytoplasm, H2A reorganizes bacterial chromosomal DNA and inhibits global transcription. Whereas bacteria recover from the pore-forming effects of LL-37, the concomitant effects of H2A and LL-37 are irrecoverable. Their combination constitutes a positive feedback loop that exponentially amplifies their antimicrobial activities, causing antimicrobial synergy. More generally, treatment with H2A and the pore-forming antibiotic polymyxin B completely eradicates bacterial growth.

[1] Department of Developmental and Cell Biology, UC Irvine, Irvine, CA 92697, USA. [2] Department of Physics & Astronomy, UC Irvine, Irvine, CA 92697, USA. [3] Department of Molecular Biology & Biochemistry, UC Irvine, Irvine, CA 92697, USA. [4] Department of Materials Sciences and Engineering, UC Irvine, Irvine, CA 92697, USA. [5] Department of Microbiology and Molecular Genetics, UC Irvine, Irvine, CA 92697, USA. [6] Cell Compartments and Signaling Group, Institut d'Investigacions Biomèdiques August Pi i Sunyer (IDIBAPS), Barcelona 08036, Spain. [7] Department of Biomedical Sciences, Faculty of Medicine, Universitat de Barcelona, Barcelona 08036, Spain. [8] Institució Catalana de Recerca i Estudis Avançats (ICREA), Barcelona 08010, Spain. ✉email: sgross@uci.edu; asirya@uci.edu

Antibiotic resistance is a worldwide epidemic. To develop new treatments, a better understanding of natural defenses may be helpful. As a first line defense, neutrophils mediate the host's response partly through establishment of neutrophil extracellular traps (NETs)[1–5]. NET formation is stimulated by virulent microorganisms that interfere with phagosomal killing, such as aggregates of pathogenic bacteria[6], including *Pseudomonas aeruginosa*[7], *Escherichia coli*[8], and *Staphylococcus aureus*[9], and fungal hyphae[10]. Histones and antimicrobial peptides have potent antimicrobial activity in NETs, but how the individual and combined effects of these components inhibit bacterial growth has not been determined[11,12].

Histones, originally proposed as antibacterial agents[13,14], are essential for NET-mediated antimicrobial activity[1]. However, extracellular histones can have toxic effects, triggering autoimmune and inflammatory responses[15], mediating mortality in sepsis[16], inducing thrombosis[17], and activating pro-inflammatory signaling through the toll-like receptors TLR2 and TLR4[18]. Thus, levels of extracellular histones must be tightly controlled.

Importantly, the histones' bacterial killing mechanism is unclear. The bulk of histone antimicrobial activity has been observed in low-ionic non-physiological solutions. At physiological magnesium levels, histones are less effective/ineffective at killing bacteria[19–25]. Since histones contribute critically to NET activity[1], but are not effective alone under physiological conditions, it is likely that histone antimicrobial activity requires coordination with other immune cell components[26].

Antimicrobial peptides (AMPs) are broad-spectrum antimicrobials[27] that co-localize with histones in NETs[1]. Many AMPs kill bacteria by forming transient pores that induce permeabilization of microbial membranes[12,28–31]. The co-localization of AMPs and histones suggests joint function. Both histones and AMPs are comparable in size, between 14–18 kDa[32,33], are cationic, contain a high proportion of hydrophobic amino acids, and possess the ability to form alpha helices. However, the ability of histones to condense mammalian DNA, a property that LL-37 lacks, raises the possibility of additional separate antimicrobial functions.

Here, we show that histone H2A and AMP LL-37 have distinct antimicrobial effects, and that together they constitute a self-amplifying, synergistic antibiotic mechanism. LL-37 forms pores that enable the entry of H2A into bacteria. H2A enhances the pores, stabilizing them and allowing entry of additional LL-37 and H2A. Once inside, H2A reorganizes bacterial chromosomal DNA, and inhibits transcription, which kills bacteria directly. Importantly, these activities are observed under physiological conditions. The combined LL-37/H2A effects are much greater than their individual effects, resulting in a synergistic antimicrobial interaction. This self-amplifying mechanism is general in nature, extending to other histones, including H3, and other AMPs, including magainin-2.

## Results

### H2A antimicrobial activity requires membrane permeabilization.

Cations stabilize the outer membrane of bacteria. We hypothesized that decreasing $Mg^{2+}$ concentrations would destabilize the bacterial membrane, increasing H2A entry and bactericidal activity. Based on free magnesium levels in human plasma and extracellular fluids[34], killing experiments were performed using two magnesium concentrations: 1 mM (physiological concentration) and 1 μM (low concentration). We assayed antimicrobial activity using 10 μg/mL histone H2A based on the finding that 15 μg/mL histones are detected in the blood plasma of baboons after *E. coli* challenge[16].

H2A treatment of *E. coli* or *S. aureus* decreased colony-forming units (CFUs) on agar plates at low magnesium (Supplementary

Fig. 1A), but not at physiological magnesium (Supplementary Fig. 1A). Similarly, in liquid cultures, H2A inhibited bacterial growth only at low magnesium (Fig. 1a, b), as measured by optical density. We note that low magnesium decreased total bacterial growth, consistent with previous reports. The inability of *S. aureus* to recover in low magnesium environments may be due to a higher sensitivity to histones in low magnesium environments.

We investigated whether H2A disrupts membranes by using propidium iodide (PI), which fluoresces upon binding nucleic acids and does not permeate the outer membranes of viable bacteria. H2A induced PI fluorescence in *E. coli* in low magnesium (Fig. 1c and Supplementary Fig. 1B), but no PI fluorescence increase was observed at physiological magnesium (Fig. 1c and Supplementary Fig. 1B), suggesting that H2A inhibits growth in low magnesium by enhanced membrane permeabilization. However, H2A-induced PI fluorescence could in principle reflect a bacterial response that induces cell death, where membrane permeabilization could be a secondary effect.

We reasoned that increased membrane destabilization due to low magnesium facilitated H2A entry. If so, membrane-permeabilizing agents could similarly increase histone entry. LL-37 is a human cathelicidin AMP that co-localizes with histones in NETs, exhibits broad-spectrum microbial activity, and disrupts lipid bilayers by forming toroidal pores[30]. LL-37 production is elevated in tissues that are exposed to microbes, such as skin and mucosal epithelia, for rapid defense against microbial infections[35]. We hypothesized that LL-37 pores could increase H2A entry.

We treated *E. coli* with LL-37 and H2A at physiological magnesium (1 mM) to avoid membrane stress from low ionic conditions. Treatment with 2 μM LL-37, a concentration reported to be the bulk minimum inhibitory concentration (MIC) of *E. coli* after 12 h[36] and a concentration below that found in inflamed epithelial cells[37], decreased the growth rate and slightly extended the lag time (Fig. 1d). H2A alone had no effect on *E. coli* growth. However, cultures treated with both H2A and LL-37 had significantly decreased growth rates compared to untreated or LL-37-treated samples. Similar effects on growth were observed using *S. aureus* (Fig. 1d), suggesting that treatment of Gram-positive or Gram-negative bacteria with LL-37 enhances the antimicrobial activity of H2A. Treatment using both H2A and LL-37 increased the PI fluorescence of *E. coli* after 1 h, indicating that increased membrane permeabilization accompanies the enhanced antimicrobial activity of H2A (Fig. 1e). Synergy is defined as an effect that is greater than the sum of each of the constituents. LL-37 and H2A are synergistic: the combined treatment inhibited growth to a larger degree than the two individual effects combined. Synergistic killing was also observed using LL-37 and histone H3 in place of H2A (Supplementary Fig. 1C), suggesting that synergy is a general property between histones and AMPs. The synergistic killing effect was diminished using citrullinated H3, which suggests histone citrullination could affect antimicrobial synergy.

Bacterial growth was not completely inhibited by treatment of LL-37 and H2A, with renewed growth observed after ~15 h (Fig. 1d). We suspect a small fraction of resistant mutants or phenotypic variants give rise to this[38,39]. The lack of complete growth inhibition was similarly observed in treatments with the bacteriostatic antibiotic chloramphenicol and bactericidal antibiotic kanamycin (Fig. 1f and Supplementary Fig. 1D), indicating that a lack of complete growth inhibition is not specific to H2A and LL-37 and may be a general property of antibiotic treatments in liquid cultures. To determine whether the combined treatment of LL-37 and H2A was bactericidal or bacteriostatic, *E. coli* were treated for 1 h and plated on agar plates that did not contain LL-37 or H2A

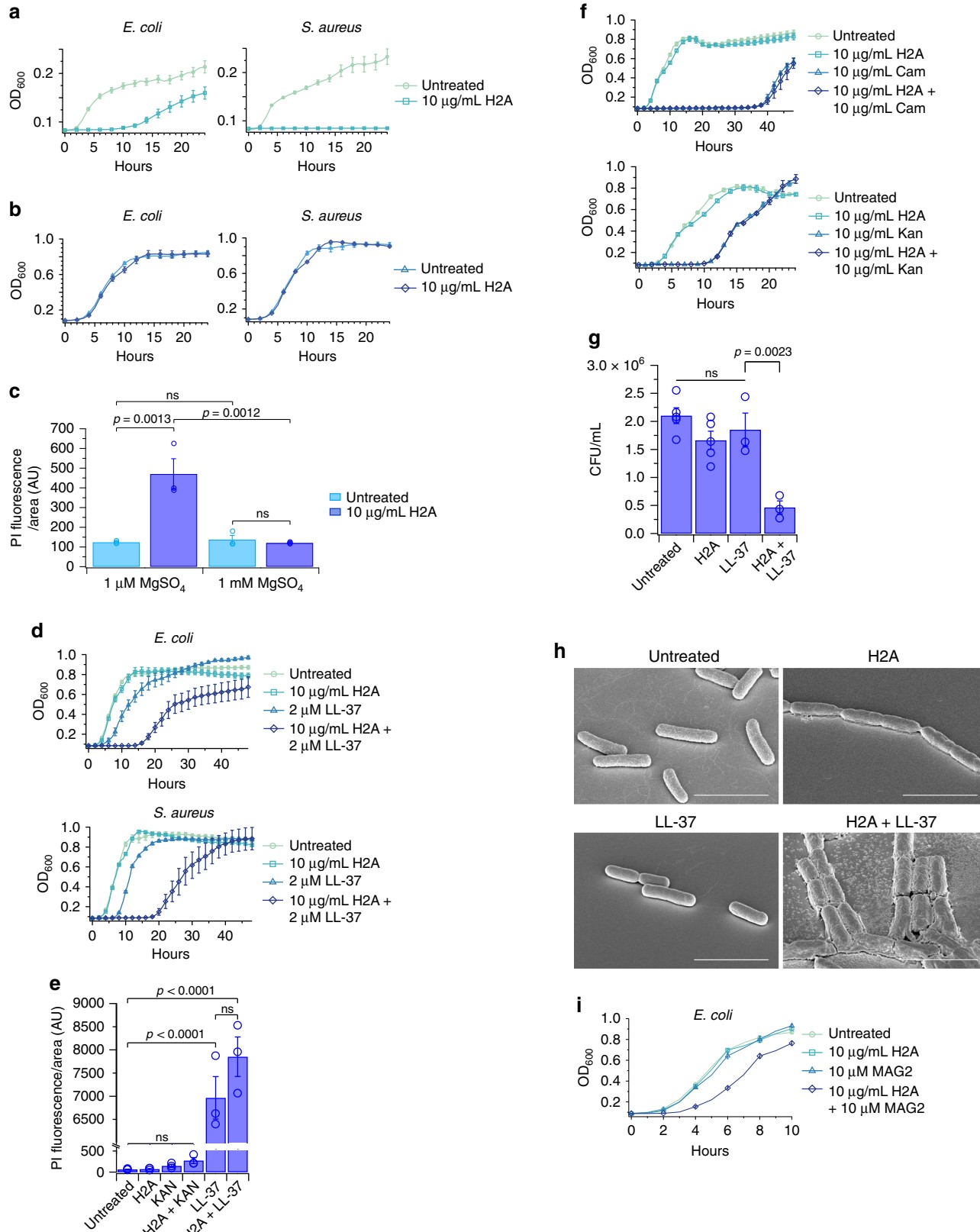

(Fig. 1g). A significant decrease in CFUs was observed, suggesting the combined H2A/LL-37 treatment is bactericidal.

The synergistic H2A/LL-37 effects were striking at the subcellular level, as measured via scanning electron microscopy (SEM) (Fig. 1h). In *E. coli* treated with either LL-37 or H2A, few cell morphological differences were observed. However, the combined LL-37/H2A treatment caused dramatic cellular damage, including cell aggregation and extensive production of insoluble components to the outer surface of the membrane and to the surrounding surfaces. In some cells, the dual treatment

**Fig. 1 Histones and the antimicrobial peptides LL-37 and magainin-2 increase killing efficacy against bacteria. a, b** Growth profiles, measured by optical density, of *E. coli* and *S. aureus* treated with H2A in media containing **a** low (1 μM) magnesium (n = 33 for each condition) and **b** physiological (1 mM) magnesium (n = 11 for each condition). **c** Intracellular propidium iodide (PI) fluorescence intensities of H2A-treated *E. coli* in 1 μM and 1 mM concentrations of magnesium after 1-h treatment (n = 3 for each condition). **d** Growth profiles of *E. coli* (n = 18 for each condition) and *S. aureus* (n = 8 for each condition) treated with 10 μg/mL H2A, 2 μM LL-37, or both in medium containing 1 mM magnesium. **e** Intracellular propidium iodide (PI) fluorescence intensities of *E. coli* treated with 10 μg/mL H2A, 2 μM LL-37, both H2A and LL-37, 10 μg/mL kanamycin (Kan), or H2A and Kan, in medium containing 1 mM magnesium (n = 3 for each condition). PI fluorescence of LL-37-treated *E. coli* and Kan-treated *E. coli* was normalized to H2A-treated cells. **f** Growth profiles of *E. coli* treated with 10 μg/mL H2A, H2A and 10 μg/mL chloramphenicol (Cam), or H2A and 10 μg/mL Kan in medium containing 1 mM magnesium (n = 6 for each condition). **g** Colony-forming units (CFU) of *E. coli* that were untreated (n = 5) or treated with 10 μg/mL H2A (n = 5), 2 μM LL-37 (n = 3), or both (n = 3) in minimal medium containing 1 mM magnesium. Bacteria were treated for 1 h before plating on non-selective LB agar plates. CFUs were normalized to H2A-treated *E. coli* CFUs in Supplementary Fig. 1A. **h** Scanning electron microscopy (SEM) images of *E. coli* treated with 10 μg/mL H2A, 1 μM LL-37, or both in medium containing 1 mM magnesium (n = 3 for each condition). **i** Growth profiles of *E. coli* treated with 10 μg/mL H2A, 10 μM MAG2, or both in medium containing 1 mM magnesium (n = 4 for each condition). Data shown as mean ± standard error of the mean (SEM) and are representative of biologically-independent experiments. One-way ANOVAs were performed. No adjustments were made for multiple comparisons. ns > 0.05. Scale bars represent 3 μm.

caused membrane bleb formation (Supplementary Fig. 1E), suggesting the LL-37/H2A combination induces membrane damage. We note that blebs were not observed in all cells and attribute this to the transient nature of the observed blebs, which is discussed below. H2A also frequently induced aggregation of cells by fusing the poles of cells together (Fig. 1h). The linkage cannot be attributed to inhibition of cell division, as aggregates contain many more cells linked together than can be duplicated through bacterial replication during the course of treatment (Supplementary Fig. 1F). Previous reports noted that positively charged molecules accumulate at the bacterial cell poles, where the Gaussian curvature is highest[40]. The bacterial aggregation here may be explained by large positive charge accumulation on the outside of the cells at the poles. LL-37 treatment did not induce aggregation and instead caused a significant reduction in cell size, consistent with induction of membrane permeabilization (Supplementary Fig. 1G).

We investigated potential synergy between H2A and the aminoglycoside kanamycin or the amphenicol chloramphenicol at physiological magnesium conditions. These antibiotics inhibit growth through protein translation inhibition[41] and cause membrane disruption[42]. Treatment of *E. coli* with either antibiotic alone increased the growth lag time (Fig. 1f). Combining H2A with either antibiotic had no additional antimicrobial effect, which suggests that these antibiotics do not synergize with H2A. Similarly, treatment using kanamycin concentration well above the MIC did not synergize with H2A (Supplementary Fig. 1D). Although aminoglycosides are reported to increase membrane permeabilization, this mechanism did not appear to form pores that enabled PI entry into *E. coli*. In addition, the combination of H2A and kanamycin had no effect on intracellular PI fluorescence of *E. coli* after a one-hour treatment (Fig. 1e). These results suggest that kanamycin and H2A do not synergize because kanamycin does not enable entry of H2A into cells. It is possible that incubation with kanamycin at higher concentrations, for longer times, or under different growth conditions would enable H2A entry into cells and produce synergy.

To determine if the H2A/LL-37 synergy was representative of a more general mechanism, we investigated the activity of H2A with the membrane-permeabilizing AMP magainin-2 (MAG2), an α-helical peptide belonging to a class of AMPs from the African claw frog (*Xenopus laevis*)[43]. Similar to LL-37, MAG2 is cationic and forms amphipathic α-helical structures in membranes. The 23-amino acid AMP forms a 2–3 nm toroidal pore, disrupting the ion gradient and inducing membrane permeabilization[44].

H2A or 10 μM MAG2 alone, a concentration below the MIC for *E. coli*, had no effect on *E. coli* growth (Fig. 1i). However,

cultures treated with both H2A and MAG2 extended the lag time significantly. Furthermore, combined H2A/MAG2 treatment significantly increased intracellular PI fluorescence after 1 h (Supplementary Fig. 1H), indicating membrane permeabilization accompanies the enhanced antimicrobial activity of H2A.

**H2A enters the cytoplasm and enhances LL-37 uptake.** We fluorescently-labeled H2A with AlexaFluor488 (AF-H2A) to track H2A localization. AF-H2A bactericidal activity was confirmed via growth inhibition assays (Supplementary Fig. 2A). Treatment of *E. coli* and *S. aureus* using AF-H2A alone produced little or no cellular fluorescence (Fig. 2a, b), indicating lack of H2A uptake, consistent with Fig. 1c, e. Under low magnesium, enhanced AF-H2A uptake was observed (Fig. 2c, d), consistent with the ability of H2A to increase membrane permeabilization in low magnesium (Fig. 1c). Treatment using kanamycin or chloramphenicol did not induce AF-H2A uptake (Fig. 2b), consistent with the inability of these antibiotics to enable the entry of H2A. Importantly, membrane permeabilization by LL-37, polymyxin B (PMB), or MAG2 significantly enhanced AF-H2A uptake (Fig. 2a, b and Supplementary Fig. 2B, C). Coupled with the growth dynamics findings above (Fig. 1d, f), these results indicate that H2A-mediated growth inhibition is concomitant with the uptake of H2A into the cell.

While H2A (at 10 μg/mL) itself does not induce membrane permeabilization at physiological magnesium (Figs. 1e and 2b), it might enhance uptake through membrane pores formed by LL-37. We thus measured effects of H2A on LL-37 uptake using fluorescently-labeled LL-37 (5-FAM-LL-37). Fluorescence was observed in the cytoplasm using a treatment of LL-37 alone (Fig. 3a, b), consistent with a previous report[45]. Importantly, H2A significantly increased LL-37 uptake over the course of 1 h (Fig. 3a–c and Supplementary Fig. 2D). Furthermore, H2A increased the localization of LL-37 to the membrane (Fig. 3c and Supplementary Fig. 2D). H2A did not shift LL-37 entirely to the membrane, as this would have been discernible as a rim pattern, such as that produced by the membrane dye FM4-64 (Fig. 3c and Supplementary Fig. 2D). Together, these results suggest H2A enhances the effect of membrane pores by enabling greater uptake of LL-37 into the cytoplasm and inducing membrane localization of LL-37.

The enhancement by H2A of LL-37 membrane pores could significantly impact ion gradients across the membrane, disrupting ATP production. We thus measured the bacterial proton motive force (PMF) using the proteorhodopsin optical proton sensor (PROPS), which increases fluorescence with a loss in PMF due to electrical depolarization[46]. The change in fluorescence is due to protonation of a Schiff base on the proteorhodopsin inside

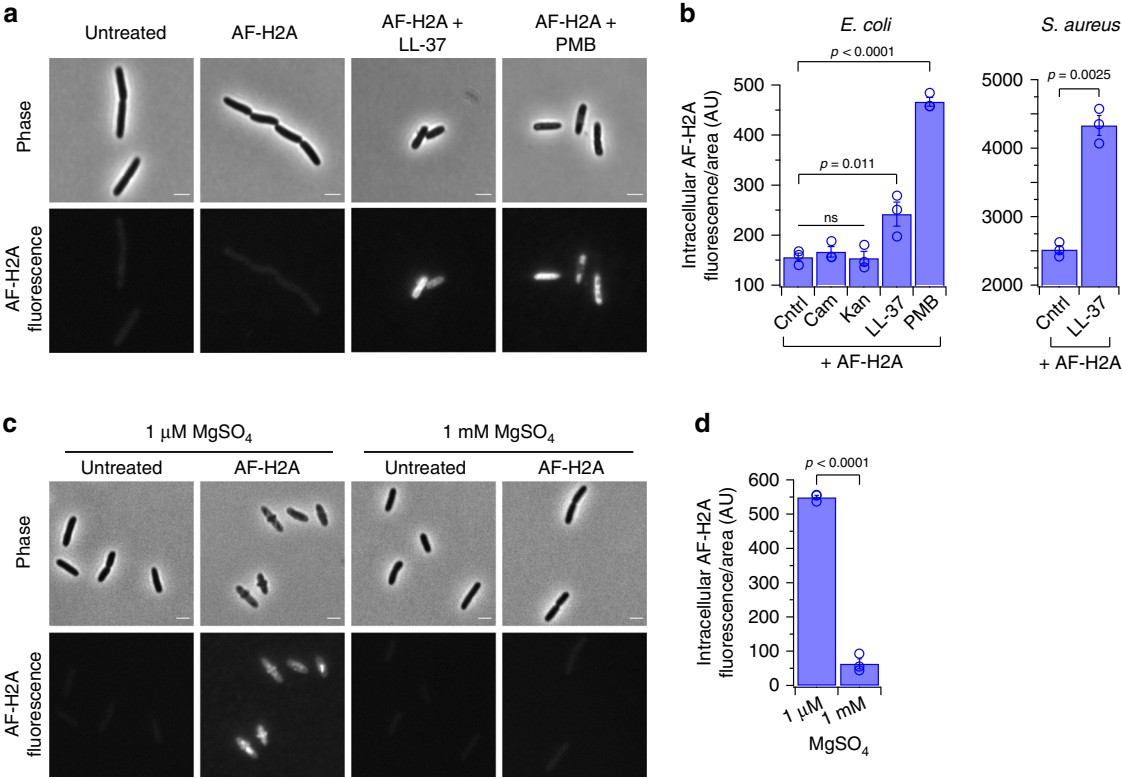

**Fig. 2 LL-37 increases the intracellular uptake of H2A. a** Fluorescence and phase-contrast images of *E. coli* that are untreated or treated with fluorescently-labeled H2A (AF-H2A) alone or in combination with 2 μM LL-37 or 1 μg/mL PMB. AF-H2A is mixed with unlabeled H2A (1% AF-H2A, combined concentration of 10 μg/mL) to decrease fluorescence intensity. The relative brightness of the PMB-treated fluorescence image was decreased for display purposes. **b** Intracellular fluorescence intensities of *E. coli* that are untreated or treated with AF-H2A alone or in combination with 10 μg/mL chloramphenicol (Cam), 50 μg/mL kanamycin (Kan), 2 μM LL-37, or 1 μg/mL polymyxin B (PMB). Intracellular fluorescence intensities of *S. aureus* treated with AF-H2A alone or in combination with 2 μM LL-37. **c** Fluorescence and phase-contrast images of untreated *E. coli* and *E. coli* treated with AF-H2A in media containing low (1 μM) magnesium and physiological (1 mM) magnesium. **d** Intracellular fluorescence intensities of *E. coli* treated with AF-H2A in media containing low (1 μM) magnesium and physiological (1 mM) magnesium. Fluorescence intensities in (**b**) were measured after a 1-h treatment with H2A alone or in combination with Cam, Kan, LL-37, or PMB. Fluorescence intensities in (**d**) were measured after a 3-h treatment with H2A. Bars indicate mean ± SEM for three independent experiments. Images are representative of three independent experiments. A one-way ANOVA was performed for the *E. coli* data in (**b**). No adjustments were made for multiple comparisons. Two-tailed and one-tailed t-tests were performed for the *S. aureus* data in (**b**) and for (**d**), respectively. Scale bars represent 2 μm.

the membrane. The fast dynamics of the reporter has captured electrical spiking in bacterial membranes due to changes in the PMF[46]. Treatment of *E. coli* with H2A had no effect on the PMF (Fig. 3d), consistent with the lack of membrane permeabilization by H2A (Fig. 1e). Cells treated with both H2A and LL-37 or with both H2A and PMB exhibited significantly higher PROPS fluorescence than cells treated with LL-37 or PMB alone, indicating that H2A further depolarizes the membrane and disrupts the PMF. We note that PROPS fluorescence is affected by the growth phase and pH[47]. All the cells were harvested at mid-exponential growth phase and were cultured in buffered medium, which minimizes potential changes in pH.

**H2A inhibits bacterial recovery and membrane repair**. We next tested bacterial recovery from H2A-induced damage. If H2A enhances uptake by membrane pores and depolarizes the membrane, these mechanisms would inhibit bacterial recovery even in the absence of AMPs and H2A. We quantified the extent of membrane repair in a strain of *E. coli* that expresses CFP under the control of a constitutively-active *ompA* promoter. *E. coli* were treated with LL-37 alone or in combination with H2A, washed to remove the treatments, and recovered in fresh medium lacking treatments. During the recovery period, cells previously treated

with H2A or LL-37 alone resumed growth and division and retained expression of CFP (Fig. 4a). In contrast, cells treated with H2A and LL-37 formed membrane blebs at the mid-cell position (Fig. 4a). Within 10 min of the formation of membrane blebs, cells lost CFP fluorescence, indicating rapid leakage of cytoplasmic contents into the surrounding medium (Fig. 4a). Blebs were observed for only 10-20 min during the recovery period (Fig. 4a), indicating that the structures were temporary. The formation of membrane blebs in dual-treated cells is consistent with the observation of membrane blebs in the SEM images (Supplementary Fig. 1E). Membrane blebs were not observed in all cells, which we attribute to the transient nature of the blebbing events.

We quantified CFP fluorescence in cells during the recovery period to measure the extent of recovery following treatment with H2A or AMPs. To ensure that the analysis focused specifically on the ability to recover, cells were also stained with propidium iodide (PI). Cells that demonstrated significant membrane damage by the treatments (PI positive) were excluded from the analysis during the recovery period. CFP fluorescence increased or remained the same during the recovery period following treatment with only LL-37 or MAG2 (Fig. 4b, c), indicating the ability of cells to recover from AMP-induced membrane pore formation. CFP fluorescence in cells treated with H2A alone was

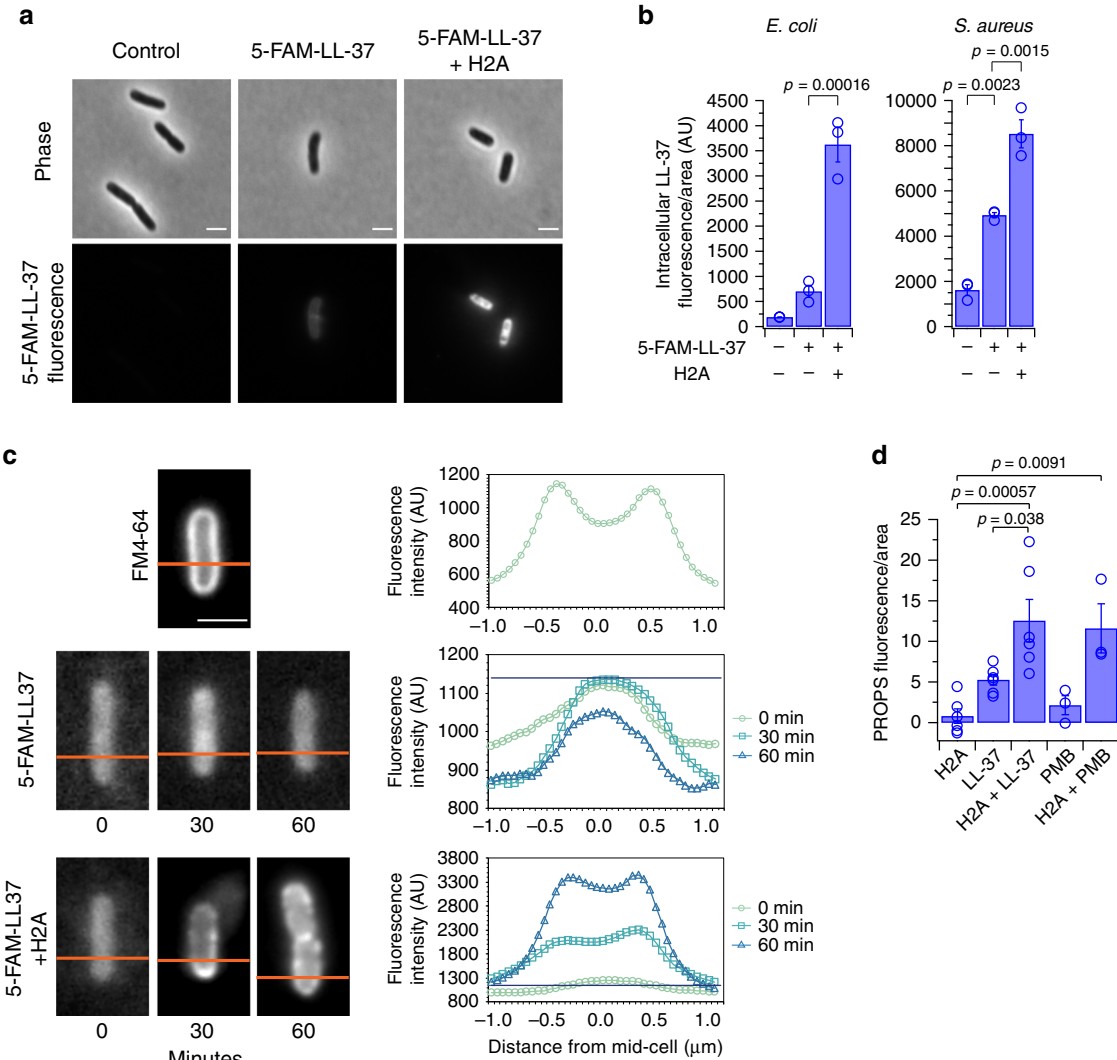

**Fig. 3 Histone H2A increases the intracellular uptake of LL-37, localizes LL-37 to the membrane, and disrupts the proton gradient with LL-37.**
**a** Fluorescence and phase-contrast images of *E. coli* that were untreated or treated with fluorescently-tagged LL-37 (5-FAM-LC-LL-37) alone or in combination with 10 μg/mL H2A. 5-FAM-LC-LL-37 is mixed with unlabeled LL-37 (1% 5-FAM-LC-LL-37, combined concentration of 2 μM) to decrease fluorescence intensity. **b** Intracellular fluorescence intensities of untreated *E. coli* and *S. aureus* or treated *E. coli* and *S. aureus* with 5-FAM-LC-LL-37 alone or in combination with 10 μg/mL H2A (*n* = 3 for each condition). Fluorescence intensities were measured after a 1-h period. **c** Representative images and associated fluorescence intensity profiles of *E. coli* that were treated with 1% 5-FAM-LC-LL-37 alone or in combination with 10 μg/mL H2A for 0, 30, or 60 min. The profiles are taken along the lines indicated in orange. The maximum fluorescence intensity of the 5-FAM-LL-37-treated cells (without H2A) is indicated by a horizontal blue line. Cell membranes were visualized using FM4-64. **d** Intracellular fluorescence intensities of *E. coli* containing the proteorhodopsin optical proton sensor (PROPS) plasmid pJMK001, which measures membrane potential. Fluorescence intensities were measured after a 1-h treatment with 10 μg/mL H2A, 1 μM LL-37, or both H2A and LL-37 (*n* = 6 for each condition); or with 1 μg/mL PMB or H2A and PMB (*n* = 3 for each condition). Bars indicate mean ± SEM of biologically-independent experiments. One-way ANOVAs were performed. No adjustments were made for multiple comparisons. Images are representative of three independent experiments. Scale bars represent 2 μm.

high across the recovery period, consistent with the lack of antimicrobial activity from H2A alone. In contrast, during recovery following a combined treatment of H2A with either LL-37 or MAG2, CFP expression remained depressed (Fig. 4b, c), indicating persistent damage from which the cell cannot recover.

The inability to recover from cell damage may reflect the specific combination of H2A with AMP-induced pores. To test this, we performed recovery experiments using low-magnesium medium in place of AMPs as an alternative method of increasing membrane permeability (Supplementary Fig. 2E) and enabling initial AF-H2A uptake (Fig. 2c, d). Treatment with H2A in the low magnesium condition significantly decreased CFP fluorescence relative to the physiological magnesium condition (Fig. 4d).

However, CFP fluorescence was restored following a 60 min recovery period and increased to levels comparable to growth in a physiological magnesium environment (Fig. 4d). Thus, H2A causes persistent cell damage that is specific to AMP-induced pores, and causes only transient cell damage when the membrane is permeabilized using a non-AMP method such as growth in low magnesium medium.

Bacterial recovery from H2A was further assessed by monitoring PI fluorescence. After 1 h of recovery, PI fluorescence in LL-37-treated cells was significantly lower compared to cells that were treated with both H2A and LL-37 (Fig. 4e). H2A-treated cells showed consistently low PI fluorescence across the recovery period, indicating the lack of membrane damage by H2A. These

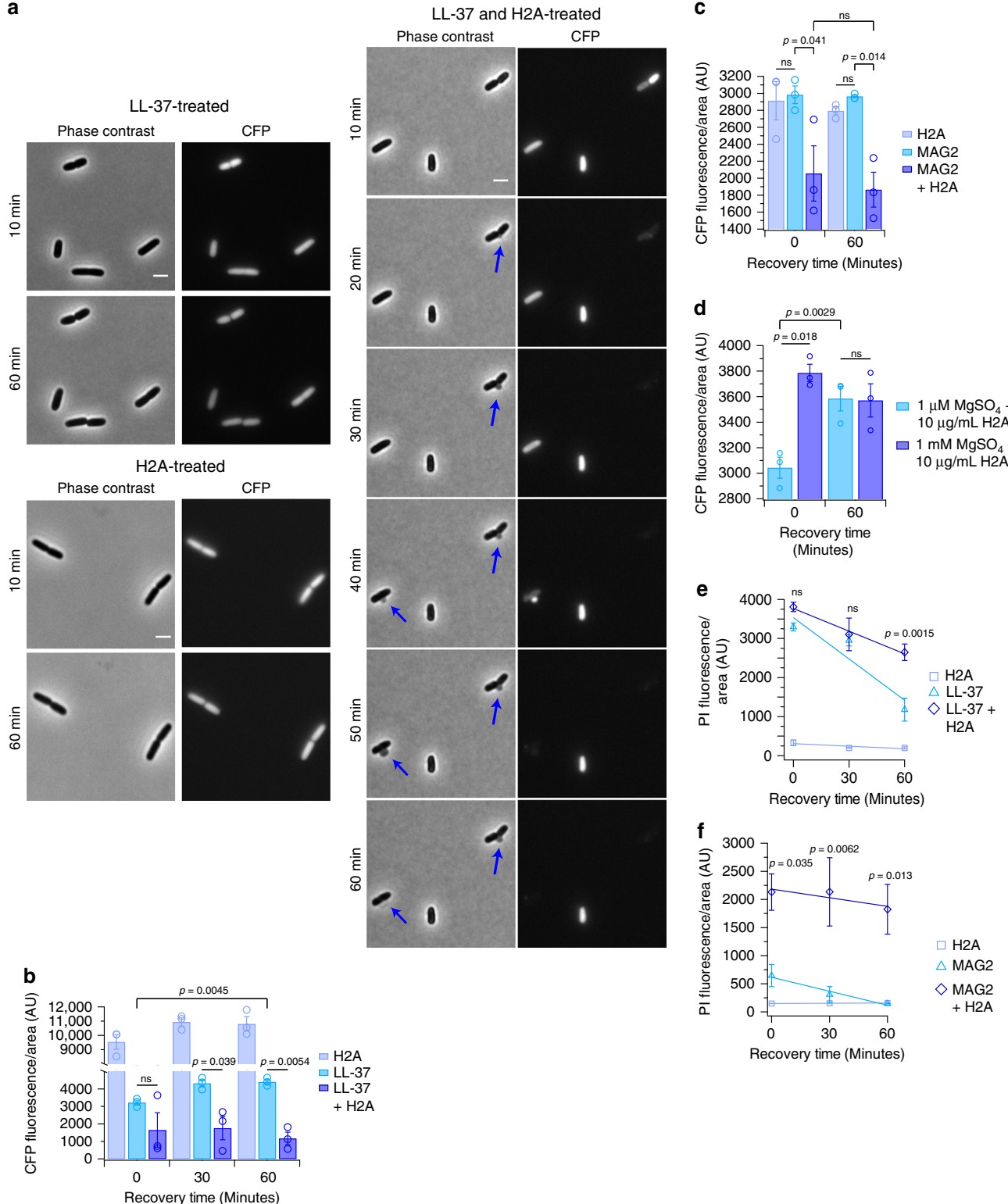

results suggest that bacteria can repair membrane pores induced by LL-37 but the presence of H2A inhibits repair of LL-37-induced pores. Similar results were observed using MAG2 in place of LL-37 (Fig. 4f). In addition, cells that were treated with H2A in low magnesium repaired membrane permeabilization within 30 min of the recovery period (Supplementary Fig. 2F), which supports the model that inhibition of repair by H2A is specific to AMP-induced pores.

Thus, H2A enhances the permeabilizing effects of AMP-induced membrane pores by facilitating H2A and LL-37 uptake, and by inhibiting repair of AMP-induced pores. We refer to this effect as pore stabilization by H2A. In further support of the pore-stabilizing effects of H2A, we observed that the dual treatment of H2A and LL-37 caused a dramatic decrease in cell size (Supplementary Fig. 2G). This result is consistent with a model in which H2A-stabilized pores enable the efflux of cytoplasmic

**Fig. 4 Histone H2A inhibits membrane recovery by stabilizing LL-37-induced pore formation. a** Fluorescence and phase-contrast time-lapse images and **b** CFP fluorescence intensities of *E. coli* that constitutively express CFP. *E. coli* were initially treated for 1 h with 10 μg/mL H2A, 1 μM LL-37, or the combination of 10 μg/mL H2A and 1 μM LL-37, and then recovered for a 1-h period without the treatments. Arrows indicate the formation of membrane blebs. **c** CFP fluorescence intensities of *E. coli* that constitutively express CFP that were initially treated for 1 h with 10 μg/mL H2A, 10 μM magainin-2 (MAG2), or the combination of 10 μg/mL H2A and 10 μM MAG2, and then recovered for a 1-h period without the treatments. **d** CFP fluorescence intensities of *E. coli* that constitutively express CFP that were initially treated for 3 h with 10 μg/mL H2A in media containing low (1 μM) magnesium and physiological (1 mM) magnesium, and then recovered for a 1-h period without the treatments. **e** Intracellular propidium iodide (PI) fluorescence of *E. coli* over a 1-h recovery following a 1-h treatment with H2A, LL-37, or the combination of H2A and LL-37. **f** Intracellular propidium iodide (PI) fluorescence of *E. coli* over a 1-h recovery following a 1-h treatment with H2A, MAG2, or the combination of H2A and MAG2. Data shown as mean ± SEM and are representative of three independent experiments. For comparison of LL-37 treatments at 0 and 60 min in (**b**), a one-way ANOVA was performed using LL-37 data only. Two-way ANOVAs were performed for all other comparisons. No adjustments were made for multiple comparisons. *p* values in **e**, **f** are indicated for comparisons between LL-37 and LL-37 + H2A and between MAG2 and MAG2 + H2A, respectively. ns > 0.05. Images are representative of three independent experiments. Scale bars represent 2 μm.

components out of the cell and disrupts the membrane. Furthermore, the result suggests a potential mechanism for the production of cellular debris in the dual-treated SEM images (Fig. 1h and Supplementary Fig. 1E).

**H2A disrupts DNA organization and suppresses transcription.** We next investigated the role of H2A subsequent to cellular entry. *E. coli* were electroporated with H2A and cultured in the continued presence of H2A. We confirmed initial entry of H2A into *E. coli* via electroporation using fluorescently-labeled AF-H2A (Supplementary Fig. 3A). The electroporation of H2A into *E. coli* had a striking inhibitory effect on growth, increasing the lag time of the cultures comparable to those treated with LL-37 alone (Fig. 5a) and decreasing CFUs to nearly undetectable levels on non-selective media following a 1 h treatment (Fig. 5b). The growth inhibition was not due to the process of electroporation or the presence of H2A alone, as these conditions had relatively minor impacts on growth (Fig. 5a, b). The dramatic decrease in CFUs on non-selective media indicate that the effects of H2A, whether through introduction into the cell by electroporation or through co-treatment with LL-37, is bactericidal and that the mechanism is rapid. Together, these results indicate that H2A has a growth-inhibitory effect in the cytoplasm and suggest that H2A affects an intracellular bacterial target.

We note that during the period that followed the electroporation of H2A, the concentration of H2A in the cytoplasm decreased in *E. coli* (Fig. 5c) despite the fact that additional H2A was present in the extracellular medium. This suggests that the presence of H2A inside the cytoplasm does not induce further membrane pore formation. In contrast, the dual treatment of LL-37 and H2A caused H2A levels to rise within the cytoplasm (Fig. 5c) and produced a striking irrecoverable effect on growth (Fig. 5a, b). This data further supports the hypothesis that H2A stabilizes the membrane pores induced by LL-37. We note that dual LL-37/H2A treatment had more growth impact than previously observed (Fig. 1d), likely due to the low magnesium present during electrocompetent cell preparation.

Although histones bind eukaryotic DNA, their ability to interact with bacterial DNA has not been characterized. We hypothesized that H2A complexes with microbial DNA, perturbing replication and transcription. To measure potential H2A-bacterial DNA interactions, we performed non-denaturing polyacrylamide gel electrophoresis of purified *E. coli* genomic DNA with H2A (Supplementary Fig. 3B, C). Increasing H2A levels inhibited DNA migration, indicating interactions between H2A and bacterial DNA (Supplementary Fig. 3B, C). LL-37 exhibited less retention of bacterial DNA for a comparable range of concentrations (Supplementary Fig. 3D, E).

We then measured the effect of H2A on *E. coli* chromosomes in live cells using Sytox Green, which fluoresces upon binding to

DNA but does not inhibit bacterial growth at low concentrations[48]. Cells were pre-treated with H2A, LL-37, or with a combination of H2A and LL-37 for 3 h. Fluorescence was distributed uniformly in untreated cells, indicating a diffuse bacterial chromosome (Fig. 5d). Treatment using H2A alone produced no change in localization, which was expected since H2A treatment does not induce H2A entry into the cytoplasm (Fig. 2a, b). Treatment using LL-37 also produced no significant change in localization, suggesting that the chromosome is not significantly perturbed by LL-37. The addition of H2A to LL-37-treated cells induced a webbed pattern in the chromosome (Fig. 5d), suggesting that H2A entry induces chromosomal reorganization. We performed principal component analysis to identify changes in the chromosomal organization over several hundred cells in each condition. This analysis provides a visualization of how similar or dissimilar chromosomal patterns are for all cells in a single condition. The clustering of cells together along the two principal components (PC1 and PC2) indicates similarities in chromosomal organization whereas those that are distributed further apart indicate dissimilarities in organization. We plotted the principal components for the four treatment conditions as normalized density plots (Fig. 5e). Most cells in untreated, H2A-treated, and LL-37-treated conditions clustered together near a single position. However, treatment using both H2A and LL-37 caused a broad distribution of chromosomal organization patterns. Importantly, this pattern of the chromosomal organization was distinct from that produced through treatment of H2A or LL-37 alone.

To further characterize H2A impact, we visualized HupA localization using a *HupA-mRuby2* construct[45]. HupA is a bacterial DNA-binding protein that reports on chromosomal organization. HupA distributions in untreated and H2A-treated cells were largely diffuse and comparable (Fig. 5f). Treatment with LL-37 condensed the fluorescence near the center of the cell (Fig. 5f), distinct from the fluorescence of untreated or H2A-treated cells (Fig. 5g). Treatment with both H2A and LL-37 caused fluorescence condensation and asymmetric localization towards the periphery of the cell (Fig. 5f), a pattern distinct from untreated, H2A-treated, or LL-37-treated cells (Fig. 5g). This result further supports the hypothesis that H2A rearranges the bacterial chromosome.

To characterize H2A's transcriptional effects, we quantified mCherry expression in an *E. coli* strain containing a tetracycline promoter (activated by anhydrotetracycline) that was transcriptionally fused to a gene encoding mCherry. Similar constructs have measured transcription in other studies[49,50]. We note that inhibition of translation could also affect mCherry fluorescence. Following 1-h pre-treatment with H2A, LL-37, or both, cells were induced for transcription using anhydrotetracycline for 1 h (Fig. 6a). Pre-treatment with H2A or LL-37 had little effect on mCherry fluorescence, indicating the H2A or LL-37 do not

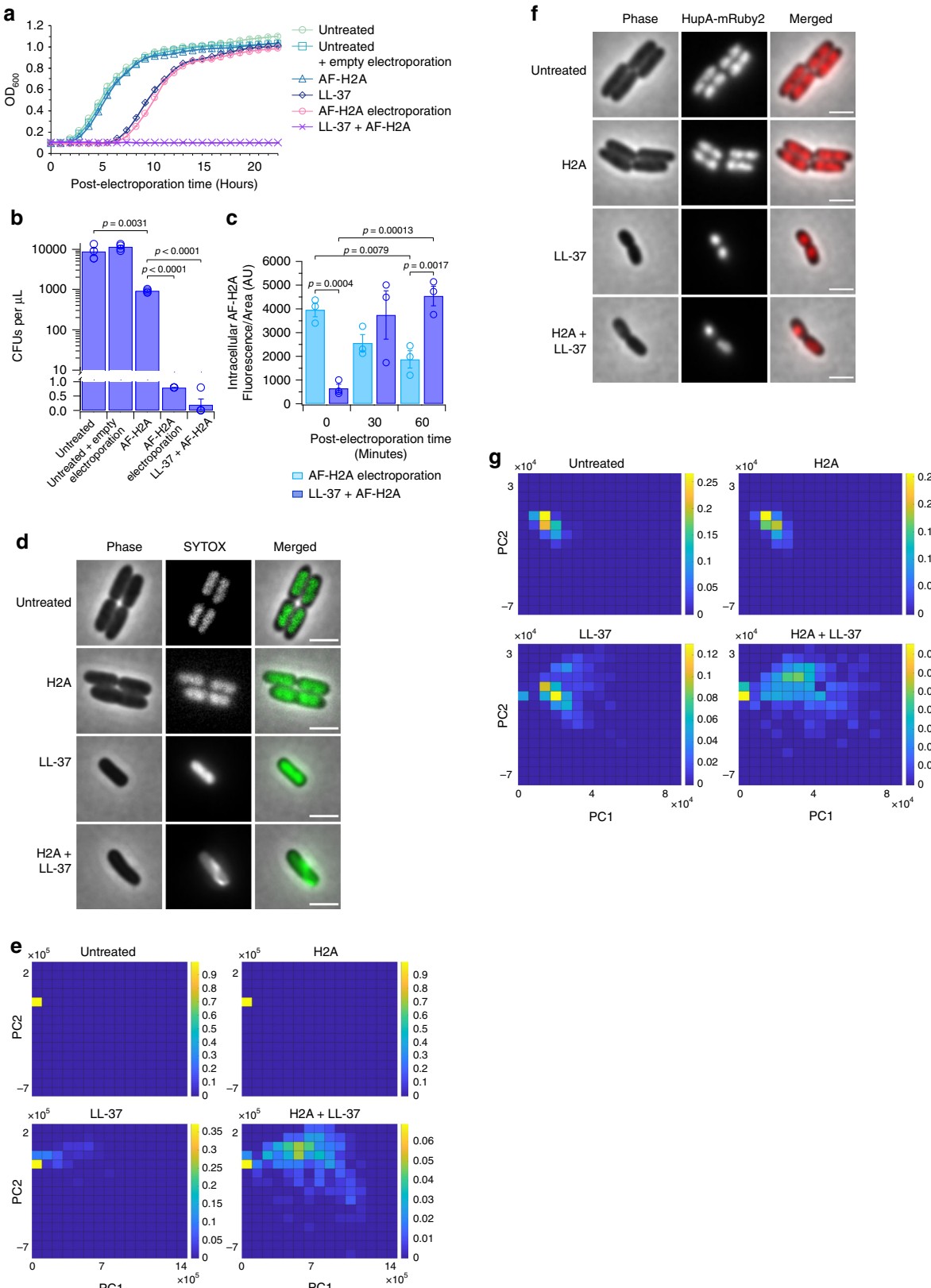

inhibit transcription induction. However, the combined H2A/LL-37 treatment decreased mCherry fluorescence, suggesting H2A inhibits transcription upon entry.

H2A's transcriptional effect was further analyzed through measurements of overall RNA production. A decrease in RNA yield was observed after 30 min of treatment with 10 μg/mL H2A and 1 μM LL-37 (Fig. 6b). Additionally, the total RNA yield was dramatically decreased after 30 min of treatment with 50 or 100 μg/mL H2A alone (Fig. 6c), which are concentrations that partially inhibited growth (Supplementary Fig. 3F). These results

**Fig. 5 H2A entry into the bacterial cytoplasm inhibits growth, perturbs chromosomal organization, and suppresses transcription. a** Growth profiles of *E. coli* that were untreated, electroporated in the absence of any treatment (empty electroporation), treated with 10 µg/mL AF-H2A, 2 µM LL-37, or both, or electroporated with 10 µg/mL AF-H2A and cultured with the same concentration of AF-H2A. Cells were cultured in minimal medium containing 1 mM magnesium. AF-H2A was mixed with unlabeled H2A (1% AF-H2A, combined concentration of 10 µg/mL) (*n* = 4 for each condition). **b** CFUs per microliter of cultures that were treated for 1 h under identical conditions as (**a**) and plated on non-selective LB plates (*n* = 4 for each condition). **c** AF-H2A fluorescence intensities of *E. coli* that were cultured under identical conditions as (**a**) (*n* = 3 for each condition). **d** Representative phase contrast, SYTOX fluorescence, and merged images of *E. coli* that were untreated or treated with 10 µg/mL H2A, 2 µM LL-37, or both. The fluorescence images are displayed using the full range of pixel values in the images. **e** Corresponding principal component analysis (PCA) of images of SYTOX-stained *E. coli* (*n* = 3 for each condition). Each cell is represented as a point on a PCA plot, which is then transformed into a density plot. The color scales indicate normalized cell densities. **f** Representative phase contrast, HupA-mRuby2 fluorescence, and merged images of *E. coli* that express HupA-mRuby2 that were untreated or treated with 10 µg/mL H2A, 2 µM LL-37, or both. **g** Corresponding PCA analysis of *E. coli* expressing HupA-mRuby2 that has been transformed to density plots (*n* = 3 for each condition). For imaging, cells were immobilized for 3 h on agarose pads containing 2 µM LL-37, 10 µg/mL H2A, or both. For SYTOX analysis, pads additionally contained 5 µM SYTOX. Bars and points are shown as mean ± SEM of biologically-independent experiments. One-way ANOVAs were performed in (**b**) and a two-way ANOVA was performed in (**c**) in which the 30-min data were excluded the analysis. See Supplementary information for raw statistical data. Images are representative of three independent experiments. Scale bars represent 2 µm.

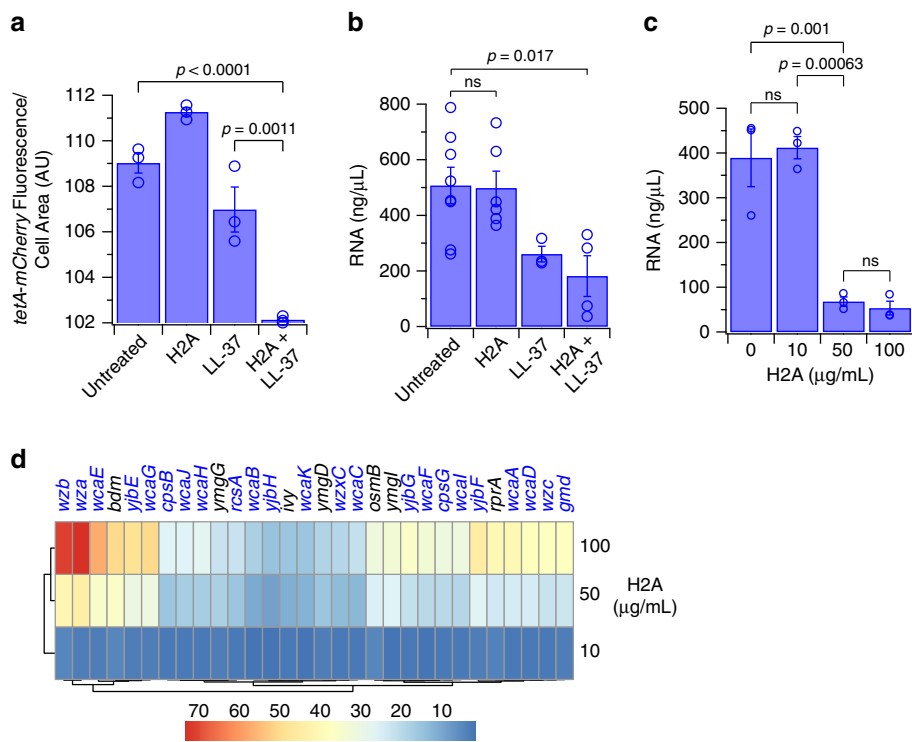

**Fig. 6 H2A suppresses global bacterial transcription and upregulates membrane biogenesis. a** Intracellular fluorescence intensities of a transcriptional reporter fusion of mCherry to a *tet*-inducible promoter. *E. coli* were pre-treated for 1 h with 10 µg/mL H2A, 2 µM LL-37, or both, and induced for transcription using anhydrotetracycline (*n* = 3 for each condition). Fluorescence was measured after 1 h. **b** RNA yields of *E. coli* that were untreated (*n* = 8) or treated with 10 µg/mL H2A (*n* = 6), 1 µM LL-37 (*n* = 3), or the combination of H2A and LL-37 (*n* = 4). **c** RNA yields of *E. coli* that were untreated or treated with 10, 50, or 100 µg/mL H2A for 30 min (*n* = 3 for each condition). **d** Top 30 upregulated *E. coli* genes in response to increasing H2A treatment, as determined through RNAseq of triplicate experiments. The majority of the genes are involved in the colonic acid/slime pathway, which synthesizes lipids and sugars that strengthen the outer membrane, are indicated in blue. Bars and points are shown as mean ± SEM and are representative of biologically-independent experiments. One-way ANOVAs were performed. No adjustments were made for multiple comparisons. ns > 0.05. Scale bars represent 2 µm.

are consistent with the ability of H2A to inhibit transcription across the population.

To understand the bacterial transcriptional response to H2A, we performed RNA-seq using increasing histone concentrations, instead of using both H2A and AMPs, since the latter condition convolves the effects of both molecules. Further, higher histone concentrations, such as 50 and 100 µg/mL H2A, may occur locally in NETs or upon release from lipid droplets.

H2A upregulated genes belonging to the colonic acid cluster, including *wza*, *wzb*, *wzc*, *wcaABCDEFGHIJKL*, *gmd*, and *wzx*, (Fig. 6d), a 19-gene cluster encoding genes for surface polysaccharide and O-antigen production[51]. Notably, these products

are recognized by host cells as hallmarks of bacterial infection and modulate pro-inflammatory responses[18]. Colonic acid also assists in maintaining the membrane potential[52]. Since H2A increases the disruption in PMF (Fig. 3d), this transcriptional response may reflect the cell attempting to repair PMF disruption. The greatest transcriptional changes induced by 10 µg/mL H2A were also the greatest changes induced by 100 µg/mL H2A (Supplementary Table 1). In particular, *wza* was upregulated 73-fold in cultures treated with 100 µg/mL H2A for 30 min. The gene cluster is tightly regulated by the *rcs* phosphorelay system, where transcription of the H-NS-regulated *rcsA* regulator increased 22-fold due to H2A treatment. We validated *rcsA* upregulation using a transcriptional

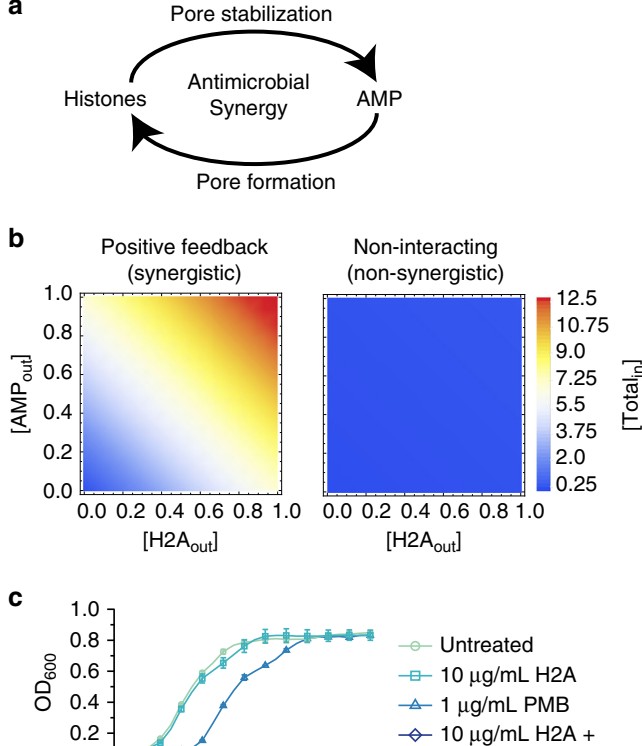

**Fig. 7 Histones and AMPs form a positive feedback loop that facilitates the uptake of antimicrobials into the cell. a** The interactions between histones and AMPs constitute a positive feedback loop in which AMP-induced pore formation increases the uptake of histones, which in turn, stabilizes the pore to facilitate uptake of additional AMPs. **b** Simulations indicating the total intracellular histone and AMP concentration for positive feedback and non-interacting relationships for a range of histone and AMP concentrations outside of the cell. Details of the simulations are described in the Methods section. **c** Growth profiles of *E. coli* treated with 10 µg/mL H2A, 1 µg/mL PMB, or both in medium containing 1 mM magnesium (n = 3 for each condition). Points are the average of biologically-independent experiments. Error bars indicate SEM.

fusion of YFP to the *rcsA* promoter in cells co-expressing CFP under control of the *ompA* promoter (Supplementary Fig. 4A). Importantly, cells that were membrane-permeabilized (PI-positive) exhibited a significant increase in *rcsA-yfp* expression and decrease in *ompA-cfp* expression (Supplementary Fig. 4B). Thus, H2A induces selective upregulation of membrane biogenesis components while globally decreasing transcription. An *rcsA* mutant showed increased sensitivity to treatment with H2A and LL-37 (Supplementary Fig. 4C), indicating its key role in promoting survival in response to H2A.

**Histone-AMP synergy due to a positive feedback loop**. Together, histones and AMPs constitute a positive feedback loop: histone entry into bacteria facilitates the uptake of AMPs, which further increases histone uptake (Fig. 7a). Feedback loops provide exponential amplification of small signals. We modeled AMP/histone dynamics using first order differential equations, encoding passive diffusion of AMPs and histones into the cytoplasm, the increase of AMP entry by histones, and the increase of histone entry by AMPs (full model details are described in the Methods). In our simulations, we observed that feedback between AMPs and

histones exponentially amplifies the uptake of both (Fig. 7b and Supplementary Fig. 5A). Thus, low concentrations of histones and AMPs together can trigger an exponential uptake of both, effectively lowering the MIC of these molecules and resulting in rapid bacterial killing. This is consistent with the all-or-none bimodal membrane permeabilization phenotype that we observed among dual-treated populations (Supplementary Fig. 5B). If interactions between the two lacked positive feedback, cells would take up far less of each (Fig. 7b and Supplementary Fig. 5A), and a continuous distribution of membrane permeabilization would instead be observed. Importantly, while bacteria can respond to a dual histone/AMP attack by increasing the expression of outer-membrane repairing machinery, this response does not defeat the exponential nature of the feedback loop and only serves to elevate the threshold concentration required to activate it.

The combination of histones, or histone fragments, with a pore-forming agent could provide a new strategy to kill bacteria. We investigated possible synergy between H2A and the cationic antibiotic polymyxin B (PMB)[53], a pore-forming antibiotic typically used as a last-resort drug. Recent reports show that polymyxin-resistant strains of *E. coli* have emerged[54]. PMB permeabilizes the bacterial membrane and enables uptake of the peptide itself[24], a mechanism similar to LL-37. Treatment of *E. coli* with 1 µg/mL PMB slightly inhibited growth (Fig. 7c). Based on our model, supplementing PMB treatment with H2A should lead to synergistic killing. Indeed, concurrent treatment of H2A and PMB completely eradicated bacterial growth (Fig. 7c) and was more effective at inhibiting growth than kanamycin or chloramphenicol at near-MIC or above-MIC concentrations using otherwise identical growth conditions (Fig. 1f and Supplementary Fig. 1D). Our model thus demonstrates that incorporating a natural defense strategy can have a potent impact on antibiotic efficacy. Identifying other synergies in natural host defenses is likely to yield important insights that can be incorporated into future antimicrobial designs.

## Discussion
The histone antimicrobial mechanism has remained elusive for decades. The work here discovers that histone H2A kills bacteria in conjunction with AMPs by inducing depolarization of the membrane potential, enhancing the effects of pore formation by AMPs, reorganizing bacterial chromosomal DNA, and repressing transcription. Importantly, the main activity of H2A is not observed in physiological environments unless a membrane pore-forming agent is present. These findings place into context the previous findings of limited histone activity in physiological conditions[19-25] and demonstrate that the antimicrobial effects of histones are unmasked when histones combine with other actors. While AMP-mediated pore formation has significant bactericidal activity, our results show that bacteria can largely recover from these effects. However, when both H2A and AMPs are present, killing is synergistic and irrecoverable.

Innate immune responses require concerted action among multiple components; the AMPs/histones activity described here represents such a mechanism. The role of AMPs in immune responses would thus appear to be the formation of membrane pores that facilitate the entry of other antimicrobial molecules such as histones into bacteria. Once inside, such molecules may target different bacterial growth mechanisms. AMPs and histones thus function as two components of a multi-step innate immunity antimicrobial mechanism.

H2A has antimicrobial activity at the membrane and within the cytoplasm. At the membrane, H2A enhances LL-37-induced pores, increasing LL-37 and H2A uptake, inhibiting repair of AMP-induced membrane pores, increasing destruction of the

gradient required for ATP production, and facilitating the release of cellular contents. Repair inhibition was not observed in membranes that were weakened through growth at low magnesium concentrations, which suggests that pore stabilization by H2A is specific to AMP-induced pores. H2A could enhance the AMP-induced pores through two mechanisms: impeding pore repair by making AMP membrane removal more difficult, or by increasing membrane tension, proposed as a mechanism of action by some AMPs[55], which would facilitate creation of AMP-induced pores and increase the difficulty of closing them.

Inside, H2A targets multiple processes providing an additional level of assault. H2A's transcriptional inhibition diminishes the bacterial response, preventing repair of cell damage caused by AMP-induced pores. However, H2A's antimicrobial activity is not fully dependent on AMP pore formation, as both growth with H2A in low magnesium conditions and the electroporation of H2A into the cytoplasm inhibited growth and were bactericidal, as judged by decreases in CFUs. We thus propose transcriptional inhibition as a potential third mechanism by which H2A stabilizes pores. We note that the electroporation of H2A inhibited growth to a less extent than the combined treatment of using both H2A and LL-37. The greater inhibition in the latter case may be attributed to the persistent pore formation by LL-37, as opposed to the transient pores formed during electroporation. Future work is required to evaluate the relative impacts of H2A acting at the membrane and within the cytoplasm on bacterial death.

H2A and LL-37 constitute a self-amplifying mechanism that significantly lowers the effective minimum inhibitory concentration of both molecules. As H2A shares significant structural and chemical similarities with other histones[33] and LL-37 shares similarities with other cathelicin-derived AMPs and defensin AMPs[32], the results and model suggest potential synergy between other histones and AMPs. In support of this, synergy was observed between histone H3 and LL-37, and between H2A and magainin-2. In certain environments, such as a lesion or NETs, proteases may cleave histones, producing histone-derived peptides[56], which could also synergize with AMPs. Importantly, as AMP antibacterial activity can be modulated, peptides may be designed in the future to increase membrane permeabilization, translocation, or synergy with other AMPs[57,58]. In principle, a single molecule that has both AMP-like and histone-like properties, such that it both induces pores in membranes and inhibits transcription, could have a self-amplifying effect by itself.

Given histone toxicity to the host[15–18], the therapeutic potential of histones and AMPs should be considered in the context of natural host defenses, including NETs and lipid droplets. The binding of histones and AMPs to NETs may prevent generalized histone spread and avoid off-target effects. Further, histone localization to lipid droplets for targeted release to kill bacteria[20] could provide an effective delivery mechanism while limiting off-target effects.

The ubiquitous co-occurrence of histones and AMPs in the immune system suggests that this antimicrobial mechanism is present in a wide range of cell types. In addition to the formation of NETs by neutrophils, histone- and AMP-rich extracellular traps form in macrophages (METs) and in dendritic cells[59,60]. Extracellular traps (ETs) have been observed in other immune cells, including mast cells and eosinophils, suggesting a role for histones and AMPs in antimicrobial activity in these cells[61]. Recent reports also demonstrate that different types of NETs are established through distinct citrullination-dependent or citrullination-independent pathways[62]. Our data suggest that the citrullinated form of histone H3 has less antimicrobial activity. The level of histone citrullination could represent a mechanism for the cell to tune the level of antimicrobial activity while balancing autoimmune activation. Future work will need to

investigate the impact of different NET formation pathways on antimicrobial activity and autoimmunity and will need to investigate the general co-occurrence of histones and pore-forming agents within and beyond the innate immunity system, where they may function effectively as a two-component antimicrobial mechanism.

## Methods

**Growth conditions.** Strains were streaked onto LB-Miller (BD Biosciences, Franklin Lakes, NJ) petri dishes containing 2% Bacto agar (BD Biosciences), incubated at 37 °C to obtain single colonies, and inoculated into MinA minimal medium[63] with 1 mM MgSO$_4$ and supplemented with 0.1% casamino acids. In summary, MinA minimal medium contains per 1 L Milli-Q water: 4.5 g KH$_2$PO$_4$, 10.5 g K$_2$HPO$_4$ 1 g (NH$_4$)$_2$SO$_4$, 0.5 g sodium citrate • 2H$_2$0, 0.2% glucose, 0.1% casamino acids, and 1 mM MgSO$_4$. Cultures were grown to stationary phase at 37 °C in a shaking incubator at 225 rpm overnight. Bacteria were either used immediately or sub-cultured to mid-exponential phase (OD$_{600}$ of 0.2). For low ionic conditions, saturated cultures growing in minimal medium containing 1 mM MgSO$_4$ were diluted 1:1000 into the minimal medium containing 1 μM MgSO$_4$.

**Bacterial strains.** Experiments were performed using the E. coli strain MG1655 (seq)[64], which is devoid of the bacteriophage lambda and F plasmid, and the S. aureus strain RN4220, which was originally derived from NCTC8325-4 was used for experiments involving Gram-positive bacteria[65]. The proton gradient was measured using pJMK001 in the E. coli strain XL1 Blue (Addagene, Watertown, MA), which expresses the PROPS protein under the control of the arabinose promoter[46]. MAL204 (MG1655 f(ompA-cfp) attλ::[P$_{rcsA}$-yfp]), constructed by Melissa Lasaro and Mark Goulian, unpublished) contains YFP fused to the promoter of rcsA, integrated at the lambda attachment site and constitutively expresses a transcriptional fusion of CFP to ompA. Chromosomal reorganization experiments were performed with a strain of E. coli containing fluorescent HupA (hupA-mRuby2-FRT-cat-FRT)[45]. MAL190 (MG1655 attλ::[cat tetR f(tetA-mCherry)], constructed by Melissa Lasaro and Mark Goulian, unpublished) contains the tetR and tetA genes integrated at the phage lambda attachment site and a transcriptional fusion of mCherry to the 3′ end of tetA. The rcsA mutant strain was constructed by P1 transduction of the D(rcsA)::kan allele from the Keio collection[66] strain JW1935 (Yale Genetic Stock Center, New Haven, CT), yielding AT14A.

**Antimicrobial peptides, proteins, and antibiotics.** Experiments involving histone treatments used calf thymus histone H2A (Sigma, St. Louis, MO), human histone H3 (Cayman Chemical, Ann Arbor, MI), or citrullinated human histone H3 (Cayman Chemical, Ann Arbor, MI). Experiments involving antimicrobial peptide treatment used the human cathelicidin LL-37 (Anaspec, Fremont, CA), FAM-LC-LL-37 (Anaspec, Fremont, CA), or magainin-2 (Anaspec, Fremont, CA). Experiments involving antibiotic treatments used kanamycin sulfate (Sigma), chloramphenicol (Sigma), or polymyxin B sulfate salt (Sigma).

**Agar plate assay.** To quantify the effects of histone treatment in low ionic conditions, overnight cultures of stationary phase E. coli or S. aureus were diluted 1:1000 into the minimal medium containing 1 μM or 1 mM MgSO$_4$ and cultured with or without 10 μg/mL histone H2A. Bacteria were cultured for 1 h at 37 °C in a shaking incubator at 225 rpm. Bacterial suspensions were diluted 1:1000 into fresh minimal medium with either 1 μM or 1 mM MgSO$_4$ and 25 μL of diluted bacterial suspension was plated on non-selective LB-Miller agar plates. Plates were incubated for 18 h at 37 °C and assessed for CFUs. To quantify the effects of synergy treatments on CFU counts, overnight cultures of stationary phase E. coli were diluted 1:1000 in minimal medium with 1 mM MgSO$_4$ and cultured with 10 μg/mL H2A, 2 μM LL-37, or both H2A and LL-37 for 1 h. After treatment, bacterial suspensions were diluted 1:1000 into fresh minimal media with 1 mM MgSO$_4$ and 25 μL of diluted bacterial suspension was plated on LB-Miller agar plates. Plates were incubated for 18 h at 37 °C and assessed for CFUs.

**Growth profiles.** Growth curve experiments were performed using a Synergy HTX multi-mode plate reader and sterile, tissue-culture treated, clear bottom, black polystyrene 96-well microplates (Corning). The temperature setpoint was 37 °C and preheated before beginning measurements. Each well contained 200 μL of total bacterial solution. For experiments performed with stationary phase bacteria, overnight cultures of bacteria were grown overnight to saturation, diluted 1:1000 into the minimal medium with 1 μM or 1 mM MgSO$_4$, and supplemented with H2A, LL-37, kanamycin, chloramphenicol, or polymyxin B. For experiments performed with exponential-phase bacteria, overnight cultures of bacteria were sub-cultured in fresh minimal medium containing 1 mM MgSO$_4$ and grown to an optical density at 600 nm (OD$_{600}$) of 0.2. Exponential-phase bacteria were diluted 1:20 into fresh minimal medium with 1 μM or 1 mM MgSO$_4$, and supplemented with H2A, LL-37, magainin-2, kanamycin, chloramphenicol, or polymyxin B. After adding antimicrobial agents, bacterial cultures were immediately added to the 96-well microplates for growth measurements. Growth curves were constructed by

taking measurements every 15 min for up to 48 h. Shaking was set to continuous orbital, with a frequency of 282 cpm (3 mm). The read speed was normal, with a 100 msec delay, and 8 measurements per data points.

**Phase contrast and fluorescence microscopy.** Fluorescence images were acquired with a Nikon Eclipse Ti-E microscope (Nikon, Melville, NY) containing a Nikon 100X Plan Apo (1.45 N.A.) objective, a 1.5X magnifier, a Sola light engine (Lumencor, Beaverton, OR), an LED-DA/FI/TX filter set (Semrock, Rochester, NY) containing a 409/493/596 dichroic and 474/27 nm and 575/25 nm filters for excitation and 525/45 nm and 641/75 nm filters for emission for visualizing the GFP and mCherry fluorescence, respectively, an LED-CFP/YFP/MCHERRY filter set (Semrock) containing a 459/526/596 dichroic and 438/24 nm and 509/22 nm filters for excitation and 482/25 nm and 544/24 nm filters for emission for visualizing CFP and YFP fluorescence, respectively, a Cy5 filter set (Chroma) containing a 640/30 nm filter for excitation, a 690/50 nm filter for emission, and a 660 nm long pass dichroic for imaging PROPS fluorescence, a Hamamatsu Orca Flash 4.0 V2 camera (Hamamatsu, Bridgewater, NJ), and an Andor DU-897 EMCCD camera. Images were acquired using Nikon NIS-Elements version 4.5 and analyzed by modifying custom-built software[67,68] (version 1.1) written in Matlab (Mathworks, Natick, MA). See 'Code Availability' section below for code. After treating bacteria with antimicrobial agents, 5 μl of culture was plated on 1% agarose-minimal medium pads and imaged immediately, which is described in ref. [69]. A minimum of 100 cells were imaged and analyzed in each experiment.

**Propidium iodide staining.** To visualize membrane permeability of stationary phase *E. coli* in low and physiological magnesium concentrations, overnight cultures of MG1655 were grown overnight to saturation and diluted 1:1000 into minimal medium with 1 μM or 1 mM $MgSO_4$, with or without 10 μg/mL H2A. 30 μM propidium iodide was co-incubated with the solution of bacteria for 1 h at 37 °C in a shaking incubator at 225 rpm before plating on 1% agarose-minimal medium pads. Data was collected using the mCherry filter. To visualize membrane lysis of mid-exponential-phase *E. coli*, overnight cultures of MG1655 were grown overnight to saturation, sub-cultured in fresh minimal medium containing 1 mM $MgSO_4$ and grown to an $OD_{600}$ of 0.2. Exponential-phase bacteria were diluted 1:20 into fresh minimal medium with 1 μM or 1 mM $MgSO_4$, and supplemented with H2A, LL-37, magainin-2, kanamycin, chloramphenicol, or polymyxin B. Bacteria were cultured at least 1 h at 37 °C in a shaking incubator at 225 rpm before plating on 1% agarose-minimal medium pads. 30 μM propidium iodide was co-incubated with the solution of bacteria for at least 15 min before imaging. Data was collected using the mCherry filter.

**SEM imaging.** MG1655 was cultured to an $OD_{600}$ of 0.2, diluted 1:20, and supplemented with 10 μg/mL H2A and/or 1 μM LL-37. Cells were treated for 1 h and added to a glass-bottomed petri dish for 15 min. Due to lower levels of adhesion, control cells were not diluted 1:20 and were incubated in the glass-bottomed petri dish for 45 min. Media was removed and 4% paraformaldehyde (PFA) was added for 20 min to fix bacteria. Dehydration was performed using serial ethanol dilutions. The fixed and dehydrated samples were coated with 10 nm of iridium using an ACE600 sputter coater (Leica Microsystems, Buffalo Grove, IL). Bacteria and surfaces were then characterized using a FEI Magellan 400 XHR Scanning Electron Microscope (FEI Company, Hillsboro, OR) at a 45° tilt angle with an acceleration voltage of 3 kV.

**Fluorescent histone labeling.** Histone 2A was fluorescently labeled with Alexa Fluor 488 NHS Ester (Invitrogen). Briefly, 10 mg of H2A was dissolved in 1 ml of 0.1 M sodium bicarbonate buffer. 50 μL Alexa Fluor dye dissolved in DMSO (10 mg/mL) was added, and the solution continuously stirred at room temperature for 1 h. A PD MidiTrap G-25 column (GE Healthcare Life Sciences, Pittsburgh, PA) was equilibrated with Milli-Q water and used to remove unreacted Alexa Fluor.

**Fluorescent histone and fluorescent LL-37 uptake.** *E. coli* strain MG1655 or *S. aureus* strain RN4220 was cultured to an $OD_{600}$ of 0.2, and diluted 1:20 into fresh minimal medium containing 1 mM $MgSO_4$. H2A uptake in *E. coli* was measured by adding 10 μg/mL AF-H2A (1% Alexa Fluor-labeled H2A mixed with 99% unlabeled H2A) and 10 μg/mL Cam, 50 μg/mL Kan, 2 μM LL-37, 1 μg/mL PMB, or 10 μM MAG2, incubating for 1 h, and analyzing using fluorescence microscopy. H2A uptake in *S. aureus* was measured using same concentrations of AF-H2A and LL-37. To measure uptake in low ionic conditions, *E. coli* was grown to $OD_{600}$ of 0.2 in minimal medium containing 1 mM $MgSO_4$, and diluted 1:20 into fresh minimal medium containing 1 μM or 1 mM $MgSO_4$. H2A uptake in *E. coli* was measured by adding 10 μg/mL AF-H2A (1% Alexa Fluor-labeled H2A mixed with 99% unlabeled H2A), incubating for 3 h, and analyzing using fluorescence microscopy. LL-37 uptake in both bacteria was measured by adding 1 μM fluorescently-labeled LL-37 (1% 5-FAM-LC-LL-37 (Anaspec) mixed with 99% unlabeled LL-37) or additionally with 10 μg/mL unlabeled H2A, incubating for 1 h, and imaging using fluorescence microscopy. Time-lapse measurements of LL-37 uptake in *E. coli* were performed by adding 2 μM fluorescently-labeled LL-37 (3% 5-FAM-LC-LL-37 mixed with 97% unlabeled LL-37) or additional 10 μg/mL unlabeled H2A, immobilizing on agarose pads, and imaging using fluorescence

microscopy. Cell membranes were visualized by adding 1.6 μM FM4-64 (MilliporeSigma, Burlington, MA), immobilizing on agarose pads, and imaging using fluorescence microscopy.

**PROPS fluorescence analysis.** The *E. coli* strain XL-1 Blue containing the PROPS plasmid pJMK001 were grown in LB in a shaking incubator at 33 °C, induced with arabinose and 5 μM retinal, and incubated in darkness for 3.5 h. The culture was spun down and resuspended in M9 minimal medium[46]. *E. coli* were back-diluted into fresh MinA minimal medium, cultured to an $OD_{600}$ of 0.2, diluted 1:20 into fresh MinA minimal medium, treated with 10 μg/mL H2A, 1 μM LL-37, both H2A and LL-37, 1 μg/mL PMB, or both H2A and PMB, and incubated for 1 h. Cells were immobilized on a 1% agarose pad and imaged using a Cy5 filter.

**Electroporation of *E. coli* with H2A.** Electrocompetent MG1655 were prepared by culturing in SOB to an $OD_{600}$ of 0.2 to 0.5, washing with 10% chilled glycerol 4 times, resuspending to an $OD_{600}$ of 0.2, and freezing at −80 °C. For electroporation, 10 μg/ml of H2A or the equivalent volume of water was added to 50 μl of electrocompetent *E. coli*, transferred to a 1 mm electroporation cuvette, and shocked using the "Ec1" setting on a Bio-Rad Micropulser (Bio-Rad, Hercules, CA). Cells were resuspended in cold MinA minimal medium with 1 mM $MgSO_4$ in a final volume of 1 mL containing 10 μg/ml of H2A, 2 μM of LL-37, or both H2A and LL-37, and cultured at 37 °C. To count CFUs, cultures were diluted serially using minimal medium containing 1 mM $MgSO_4$ and 25 μL of the dilutions were plated on non-selective LB-Miller agar plates. Plates were incubated for 18 h at 37 °C and assessed for CFUs by counting the number of colonies present.

**Timelapse of *E. coli* recovery.** To quantify the time-course of recovery in *E. coli* treated with H2A alone, LL-37 alone, or the synergistic combination of LL-37 and H2A, MAL204 was cultured to mid-exponential phase in MinA minimal medium, treated with 10 μg/mL H2A, 1 μM LL-37, or 1 μM LL-37 with 10 μg/mL H2A, and incubated for 1 h. The solution was filtered through a 0.22 μm filter to remove excess LL-37 and H2A and cells were resuspended in fresh minimal medium. Cells were immobilized on a 1% agarose pad and imaged over an hour time period.

**Time course of membrane healing.** To quantify the time-course of membrane repair in bacteria treated with H2A alone, AMPs alone, or the synergistic combination of AMPs and H2A, MAL204 was grown to mid-exponential phase in MinA minimal medium, diluted 1:20 with 10 μg/mL H2A, 1 μM LL-37, 1 μM LL-37 with 10 μg/mL H2A, 10 μM MAG2, or 10 μM MAG2 with 10 μg/mL H2A, and incubated for 1 h. The solution was filtered through a 0.22 μm filter to remove excess LL-37 and H2A and cells were resuspended in fresh minimal medium. Cells were allowed to recover for 0, 30, and 60 min before the addition of 30 μM propidium iodide for 15 min prior to performing fluorescence microscopy. Intracellular propidium iodide fluorescence and CFP fluorescence were quantified. To quantify the time-course of membrane repair in bacteria treated with H2A in low and physiological environments, MAL204 was grown to mid-exponential phase, diluted 1:20 into minimal media with 1 μM or 1 mM $MgSO_4$, with or without 10 μg/mL H2A, and incubated for 3 h. The solution was filtered through a 0.22 μm filter to remove excess H2A and cells were resuspended in fresh minimal medium. Cells were allowed to recover for up to 60 min before the addition of 30 μM propidium iodide for 15 min prior to performing fluorescence microscopy. Intracellular propidium iodide fluorescence and CFP fluorescence were quantified.

**Cell aggregate size and cell size analysis.** MG1655 were cultured to an $OD_{600}$ of 0.2, diluted 1:20 into fresh MinA minimal medium, treated with 0-4 μM LL-37 or 0-100 μg/mL H2A, and incubated for 1 h. Cells were immobilized on an agarose pad, imaged using phase contrast microscopy, and analyzed using our own custom-written image analysis tools[67,68] version 1.1 that was written in in Matlab (Version R2017b; Mathworks, Natick, MA). See the "Code availability" section below to download code. The total pixel area of each individual cell was determined by computing the mask area and converting from pixels to $\mu m^2$ by multiplying the mask area by a factor of 0.00422 $\mu m^2$/pixel to account for the microscope camera pixel size and objective magnification.

**Chromosomal analysis using SYTOX and HupA-mRuby2.** MG1655 or XL-1 Blue expressing HupA-mRuby2 was cultured to an $OD_{600}$ of 0.2, diluted 1:20 into fresh MinA minimal medium, and treated with 2 μM LL-37, 10 μg/mL H2A or both LL-37 and H2A for 30 min. For SYTOX visualization, MG1655 were stained with 3 μM SYTOX Green nucleic acid stain (ThermoFisher, Waltham, MA) for 10 min. Cells were immobilized on agarose pads containing 2 μM LL-37, 10 μg/mL H2A, or both LL-37 and H2A, and remained on the pad for 3 h before imaging using the fluorescence microscopy. For SYTOX analysis, pads additionally contained 5 μM SYTOX Green. Raw images were analyzed through principal component analysis using our own custom-written image analysis tools[67,68] version 1.1 and modified in Matlab (Version R2017b; Mathworks, Natick, MA). See 'Code Availability' section below for code. Individual cells were identified in phase-contrast images using canny edge detection and using SuperSegger[70]. Images of LL-37-treated cells and of cells treated with both LL-37 and H2A were pooled

together, rotated such that the major axis of the cell was parallel to the *x*-axis and resized to 30 × 100 pixels. The covariances between corresponding pixels of different cells were computed using the 16 bit intensity values from the rotated and resized fluorescence images and for the same images rotated by an additional 180 degrees. The orientation that gave the lower covariance was used for the analysis. Principal components for the covariance matrix were computed using ~400 cells and the principal components that gave the two largest eigenvalues were plotted. Density plots were created by binning points in principal component space in a 15 × 15 bivariate histogram plot. The size of each bin was determined by subtracting the minimum principal component score from the maximum principal component score and dividing that by the number of bins along that principal component. Histogram bins were normalized as the fraction of the total cell population.

**Bacterial DNA purification**. Overnight MG1655 cultures were grown to saturation in MinA minimal media. DNA purification was performed using a Miniprep kit (Qiagen, Germantown, MD). A three-second sonication step was performed after lysis to isolate genomic DNA.

**Non-denaturing nucleic acid PAGE**. 10 μL mixtures containing 1 μg purified DNA from MG1655 were incubated with 0–1.4 μg Histone H2A or LL-37 for 25 min at 25 °C. Gel loading sample buffer (5×, Bio-Rad, Hercules, CA) was added to a final concentration of 1× and the products were separated by native PAGE on a 5% TBE gel (Bio-Rad, Hercules, CA) at 100 V for 60 min. The gel was stained with 1X SYBR safe (Invitrogen, Carlsbad, CA) in TBE buffer[71] for 30 min before visualization using a EOS Rebel T5 DSLR camera with an f/3.5–5.6 18–55 mm lens (Canon, Huntington, NY) and a DR46B Transilluminator (Clare Chemical, Dolores, CO).

**In vivo transcription assay**. To determine how histone entry into the bacterial cell impacts transcription, MAL190 was cultured to mid-exponential phase, diluted 1:20 in MinA minimal medium, treated with 10 μg/mL Histone H2A and/or 2 μM LL-37, incubated for 1 h, and induced for transcription using 50 ng/mL of anhydrotetracycline. The fluorescence of mCherry was measured after 1 h using fluorescence microscopy.

**RNAseq**. MG1655 were cultured to saturation overnight in MinA minimal medium, diluted 1:1000 into the same medium, cultured to an OD$_{600}$ of 0.2, diluted 1:20 in pre-warmed medium, and supplemented with 10 μg/mL H2A, 1 μM LL-37, or both. 10 mL of culture was harvested at 0, 30, and 60 min, filtered through a 0.8 μm filter, washed with 2 mL H$_2$O, and resuspended in 600 μL Total Lysis Solution (TE 8.0 (10 mM Tris-HCl, 1 mM EDTA), 0.5 mg/mL lysozyme (Sigma), and 1% SDS). Samples were incubated for 3 min at room temperature before snap freezing in liquid N$_2$. Samples were kept in −80 °C until nucleic acid extraction with a hot phenol-chloroform extraction and ethanol precipitation[67]. RNA yield was measured using a Nanodrop 2000 (Thermo Fisher, Waltham, MA). Samples were digested with DNase (Ambion, Waltham, MA) and treated with RiboZero (Illumina, San Diego, CA). A NEBNext Ultra Directional Library kit (NEB, Ipswich, MA) was used to construct a cDNA library, which was sequenced by the Princeton University Genomics Core Facility with a depth of at least 10 M read per experimental condition. Sequencing data were analyzed using our own software written in Python version 2.7.16 and using R version 3.4.3 (The R Foundation, Vienna, Austria). Sequences were aligned to the MG1655 genome (U00096.3) using Bowtie2[72] version 2.2.4.

**Transcripton of *rcsA***. MAL204, which contains YFP fused to the promoter of *rcsA* and constitutively expresses a transcriptional fusion of CFP to *ompA* was grown to mid-exponential phase. Bacteria were diluted into warmed MinA minimal medium with increasing concentrations of H2A. In addition, 30 μM PI was added to the culture to specifically measure fluorescence intensities in membrane-permeabilized cells. After a 30-minute incubation period, cells were immobilized on a 1% agarose pad and YFP, CFP, and PI fluorescence values were analyzed using fluorescence microscopy.

**Statistical analysis**. Statistical analysis was performed by running Welsh t-tests or ANOVA and Tukey's post-hoc tests using R 3.4.3 (Kite Eating Tree), Image J (v1.51k), Microsoft Excel version 16.36, or our own custom-written MATLAB scripts version 1.1. See the "Code availability" section below for code.

**Histone-AMP positive feedback model**. We developed a mathematical model to describe the dynamics of histone and AMP uptake into bacterial cells. Histones and AMPs enter passively using simple diffusion:

$$\frac{d[\text{His}_{in}]}{dt} = k_{\text{Hisentry}}[\text{His}_{out}] \quad \text{and} \quad \frac{d[\text{AMP}_{in}]}{dt} = k_{\text{AMPentry}}[\text{AMP}_{out}]$$

where [His$_{in}$] and [His$_{out}$] represent the concentrations of histones inside and outside of the cell, respectively, [AMP$_{in}$] and [AMP$_{out}$] represent the concentrations of AMP inside and outside of the cell, respectively, and $k_{\text{Hisentry}}$ and $k_{\text{AMPentry}}$

are the rate constants associated with the passive entry of histones and AMPs into the cell, respectively. Molecules of histones and AMPs can leave the cell through a number of ways including cell division, shedding of cell components, and transport through drug efflux pumps. We describe these combined effects on histones and AMPs using the rate constants $k_{\text{Hisexit}}$ and $k_{\text{AMPexit}}$, respectively. To encode the behaviors that histones increase the intracellular AMP concentration and that AMPs increase intracellular histone concentrations, potentially through pore-stabilization, we defined the rate constants $k_{\text{Hisstab}}$ and $k_{\text{AMPstab}}$, arriving at the equations:

$$\frac{d[\text{His}_{in}]}{dt} = k_{\text{Hisentry}}[\text{His}_{out}] - k_{\text{Hisexit}}[\text{His}_{in}] + k_{\text{Hisstab}}[\text{AMP}_{in}]$$

$$\frac{d[\text{AMP}_{in}]}{dt} = k_{\text{AMPentry}}[\text{AMP}_{out}] - k_{\text{AMPexit}}[\text{AMP}_{in}] + k_{\text{AMPstab}}[\text{His}_{in}]$$

In our simulations, we set the initial histones and AMP concentrations inside the cell to 0. The concentration of histones and AMPs outside of the cell remained constant, which describes an environment in which there is an excess of histones and AMPs. We set the permeation rates of $k_{\text{Hisentry}}$ and $k_{\text{AMPentry}}$ to 0.004 s$^{-1}$ based on permeation measurements of the charged antibiotic tetracycline into bacterial cells[73]. The rate constants $k_{\text{Hisexit}}$ and $k_{\text{AMPexit}}$ were set to correspond to a doubling time of 30 min, which is a conservative estimate of the rate of histone and AMP removal from the cell that does not require the existence of an export mechanism. We simulated the synergy condition by setting $k_{\text{Hisstab}}$ and $k_{\text{AMPstab}}$ to 0.1 s$^{-1}$ and simulated the non-synergistic condition by setting these rate constants to 0 s$^{-1}$. For the uptake dynamics figure, we set the concentrations of histones and AMP outside of the cell to 1 and computed the total intracellular concentration of these molecules as a function of time. Density plots were constructed by computing the total intracellular concentration of histones and AMPs following 60 min of exposure to a range of histones and AMPs concentrations outside of the cell.

**Reporting summary**. Further information on research design is available in the Nature Research Reporting Summary linked to this article.

## Data availability

Raw data for all figures that contain ANOVA or *t*-test analyses are available in the Supplementary Information Source Data file. The RNA-Seq data is freely available under the National Center for Biotechnology Information Gene Expression Omnibus accession number GSE142755. Additional raw data that support the findings of this study are available from the corresponding authors upon request. Source data are provided with this paper.

## Code availability

The custom MATLAB scripts used for processing and analyzing the fluorescence microscopy data, and the custom Python scripts (for Python version 2.7.16) used for RNA-Seq are freely available as package version 1.1 from Zenodo at [https://doi.org/10.5281/zenodo.3898289].

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

## Acknowledgements

We thank D. Mauzy-Melitz, E. Pearlman, and M. McClelland for helpful discussions, S. Taheri-Araghi for the HupA-mRuby2 strain, M. Lasaro and M. Goulian for *E. coli* strains, and A. Trinh for construction of the *rcsA* mutant strain. Sputter coating and SEM were performed at the Irvine Materials Research Institute (IMRI) at UC Irvine. PROPS cells were obtained from the Lawrence Berkeley National Laboratory Molecular Foundry. Work at the Molecular Foundry was supported by the Office of Science, Office of Basic Energy Sciences, of the U.S. Department of Energy under Contract No. DE-AC02-05CH11231. S.G. was supported by Human Frontiers grant HFSP RPG0020, T.D. was supported by the Department of Education GAANN fellowship, and A.S. was supported by NIH R21 Grant R21AI139968.

## Author contributions

T.D., H.A., L.D., and L.U. performed the experiments and analyzed data. T.D. wrote the initial draft of the paper. R.R. performed SEM imaging. T.D., H.A., L.D., L.U., S.G., and A.S. designed experiments, discussed results, and edited the paper. M.B. and A. P. discussed results and edited the paper.

## Competing interests

The authors declare no competing interests.
