## [Peer Review File · Nature Communications]

Reviewers' comments:

Reviewer #1 (Remarks to the Author):

This manuscript examines the antimicrobial activity of histones, which are now better known for their ability to organize DNA in mammalian chromosomes. Specifically, the authors investigate how histones can work together with membrane permeating antimicrobial peptides such as LL37 to achieve synergistic antimicrobial effects, such as disruption of PMF and DNA organization.

This paper engages a long standing problem and makes exciting progress. In particular, the concept of synergy (such as that between different types of AMPs) have been proposed, but the demonstrations are not as nice as that shown here. I do have a number of questions, comments, and suggestions, which I hope the authors can take into consideration.

This observation does not disagree or detract from the conclusions in this paper, as histones can contribute to antimicrobial activity in multiple ways. However, it is known that fragments of histones can be more antimicrobial than complete histones. Histone fragments, often the N-terminal fragment, where membrane permeation activity resides, may be more active (and more active at elevated salt levels). In a heterogeneous environment such as a lesion or the immediate vicinity of NETs, there may be proteases that can create such histone fragments. It may be a good idea for the authors to mention this just to cover the possibility.

Lines 124 – 128: The authors go through a chain of logic whereby they show that unlike LL37, antibiotics that act via protein synthesis inhibition but do not permeate membranes are not synergistic with H2A, which does not permeate membranes under low divalent conditions, therefore these observations indicate that the antimicrobial effect of H2A requires membrane permeabilization. At this point of the paper, perhaps the authors mean 'consistent with' and not 'indicating'? strictly speaking, the conclusion is not possible until the localization studies of H2A done in the next section.

The use of PROPS to monitor PMF is a nice touch. It will be helpful to the reader to point out potential limitations of this technique for the present purpose.

There are several points in the paper that can benefit from more clarifying text: For example, the PCA for the induced chromosome patterns, and a fuller discussion of the feedback loop model in Fig 5, so that the reader can better evaluate assumptions etc.

From the conclusions of the present paper, can we then say that an AMP like buforin, which is derived from H2A and can permeate membranes, self-amplifying?

Reviewer #2 (Remarks to the Author):

The manuscript by Doolin et al. focuses on the synergistic anti-microbial effects of histone 2A and the AMP LL37 in vitro. It provides a very interesting, novel concept for immune defence and covers different aspects of this mechanism. However, some important questions remain unanswered and some of the methods should be improved for what the authors are trying to show.

General concerns:

- All experiments were conducted with H2A. Would the other histones be expected to have similar effects? This should at least be discussed.
- The authors mention NET formation as a mechanism where histones have an anti-microbial role. A hallmark of NET formation is global citrullination of histones. Would this alter the here-shown mechanism? It would be good to investigate this, especially if the authors discuss NET formation as a main source of histones and AMPs.

Also, other AMPs apart from LL37 should be considered. They are only very generally discussed.

- How do the bacteria recover from LL37-H2A treatment? The growth curves indicate bacterial growth after approx. 15-18 hours -how is this possible? This should at least be discussed.
- The proposed mechanism of pore stabilization is not at all clear. Is there a direct interaction between the molecules? In my eyes, this is the main weakness of the paper.
- Chromatin rearrangement shown by Sytox Green staining is not convincing.
- The importance of alterations of transcription (and translation?) at high concentrations of H2A remains unclear in relation to the proposed pore-formation action.
- In vivo toxicity of histones should be considered in the introduction and discussion.

In more detail:

Introduction:

- As mentioned above, other functions of histones in vivo and in the immune system should be discussed, such as the deleterious effects mediated through TLR2 and 4 (Allam et al., 2012, J Am Soc Nephrol., Xu et al., 2011, J Immunol.,) as well as their prominent role in sepsis (Xu et al., 2009, Nat Med). Also the role of histones in thrombosis should be considered (in the introduction or the discussion).

Results:

- PI staining of bacteria incubated with H2A under low magnesium conditions: PI-positivity could also be the result of the bacteria undergoing apoptosis or of the membrane becoming more permeable as a secondary effect. How does this experiment show that H2A is actually inhibiting growth by primarily increasing cell membrane permeability, as the authors suggest?

- Figure 1C, figure legend: It would be helpful to indicate at what time point PI-fluorescence was measured in the legend. Generally, more information on treatment times in the figure legends would be good.

- Figure 1A: E. coli appears to recover and proliferate after an initial inhibition, unlike S. aureus. How can this be explained? Is this difference attributable to the gram-status of the bacteria?

- In Figure 1E I find the way the bacteria are attached to each other after H2A treatment very intriguing, however I did not understand how this would happen. Is there any way to explain this very specific morphology? Especially in light of figure 2, where the authors show that H2A hardly enters the cell without formation of pores. How can this be explained?

- In Figure 1E, it might be helpful to show lower power magnifications of the bacterial cultures as here it looks like there are more bacteria in the picture with H2A plus LL37 – I assume this is just a clustering effect, though?

Also, please check Figure numbering – there is no Figure 1G, for example but it is mentioned in the text.

Figure 1D:

- While there is an impressive delay in bacterial growth, after about 20 hours bacteria appear to be proliferating quite well again – do they develop some kind of resistance mechanism or how is the pore formation overcome?

Figure 2A:

- Fluorescence of bacteria treated with H2A alone should also be shown.

To claim that H2A enhances/facilitates pore formation by LL37 I don't think it is sufficient to measure cytoplasmic fluorescence of labelled LL37. Rather, it would be very helpful to actually show membrane localization of LL37, even if only in a semi-quantitative way, by higher resolution techniques.

- How would H2A stabilize LL37 pores? Would the two proteins interact directly, both of them being cationic? Has this been shown or can this be modelled? As the authors are stressing this synergistic action very strongly, more information on how this would occur needs to be provided. Would this interaction not change upon citrullination of histones in NET formation?

Figure 4B:

- The "reorganization of the chromatin" is not convincing. How do the authors know this is truly a reorganization and not just steric/static interference of H2A with the binding of Sytox Green?

- Also, both the text in the Results section and the figure legend should mention for how long the cells were treated with H2A before the principal component analysis was performed.

- Finally, is there a way to perform a more functional analysis of the chromatin – as far as I understand, the principal component analysis mainly focuses on the staining pattern of the chromatin? Again, changes in this analysis might as well show altered staining behaviour instead of true reorganization of the chromatin.

Figures 4D and E:

- How is mCherry fluorescence a specific indicator of transcription? A decrease in fluorescence could also indicate less translation, could it not?

- In Figure 4E, much higher concentrations of H2A than in the other experiments were needed to achieve an effect. Are these concentrations likely to be reached under physiological conditions? Do the authors think this is a relevant mechanism of action, if the other effects (pore stabilization, membrane depolarization) are seen at much lower concentrations?

- Why was RNA synthesis not investigated after treatment with LL37 plus H2A? As all other experiments were conducted in this fashion, this should be added.

Discussion:

- The authors formulate a general defense mechanism for AMPs together with H2A. Other AMPs with similar effects should at least be discussed, if not shown in a proof of principle experiment.

- As mentioned above, histones mediate significant toxicity in vivo, leading to organ damage, sepsis, thrombosis etc. This should be discussed here if application of histones is proposed as an antimicrobial therapy. Also, it should be considered that binding of histones and AMPs to NETs may be a mechanism to prevent generalized spread of histones (anti-inflammatory mechanism of aggregated NETs).

Minor:

In the methods section, the authors sometimes use μl and sometimes ul .

There is no figure 1G, although it is mentioned in the text.

Line 257: delete the "a".

Reviewer #3 (Remarks to the Author):

In this article, Doolin and collaborators explore the interactions of the histone H2A and the antimicrobial peptide LL-37. Co-treatment of *E. coli* with these two cationic proteins potentiates delayed bacterial growth, increased uptake of PI, and membrane blub formation. The results are expected since both antimicrobial proteins are well documented to act on the cytoplasmic membrane. The manuscript was hard to follow. Some of the supplementary figures were not commented, and main figures were out of order. Important experimental details were not included, making it difficult to interpret the data. Some of my concerns are:

Fig. S1A is not commented in main text. Similar analyses should be presented as measure of antimicrobial activity of LL-32 and H2A. Figure 1E is wrongly referenced. Some of the primary data are shown for the first time in Discussion.

Fig 1G is missing. SEM images don't clearly show membrane disruption. It is surprising that membrane blubs are not seen by SEM, when they seem to be formed in the bright fields.

Not clear how cell size was calculated.

Increased antimicrobial activity of H2A in low magnesium is likely due to disruption of outer membrane and increased uptake of H2A into periplasm. This should be tested.

The experiments with kanamycin (50 ug/ml!) are hard to interpret. At these high concentrations kanamycin causes death. The experiments should be done with doses closer to MIC. By the way, aminoglycosides cause membrane disruption through mistranslation of proteins. If the proposed model is correct, one would expect synergism between Km and H2A.

Experiments in Figure 3 and 4 are missing important controls. What are the phenotypes of cells treated with LL-37 or H2A by themselves?

The authors claim that H2A stabilizes pores formed by LL-37. However, the data do not conclusively show such a mode of action. It is equally possible independent pores by both antimicrobial peptides work together.

Most antimicrobial activity emanating from cotreatment of E. coli with LL37 and H2A appears to be manifested in the form of bacteriostasis, raising concerns about therapeutic potential as claim in the abstract.

Figure 4A is unnecessary, since it is well known histones bind to DNA. The data could be moved to supplementary information.

Line 201, Principal component analysis should be explained for people unfamiliar with the procedure.

We thank the reviewers for their interest in this work and for their helpful suggestions and comments. These comments and suggestions have been addressed as described below.

Reviewer comment are in **black**.

Author responses are in **blue**.

Reviewer #1 (Remarks to the Author):

This manuscript examines the antimicrobial activity of histones, which are now better known for their ability to organize DNA in mammalian chromosomes. Specifically, the authors investigate how histones can work together with membrane permeating antimicrobial peptides such as LL37 to achieve synergistic antimicrobial effects, such as disruption of PMF and DNA organization.

This paper engages a long standing problem and makes exciting progress. In particular, the concept of synergy (such as that between different types of AMPs) have been proposed, but the demonstrations are not as nice as that shown here. I do have a number of questions, comments, and suggestions, which I hope the authors can take into consideration.

1. This observation does not disagree or detract from the conclusions in this paper, as histones can contribute to antimicrobial activity in multiple ways. However, it is known that fragments of histones can be more antimicrobial than complete histones. Histone fragments, often the N-terminal fragment, where membrane permeation activity resides, may be more active (and more active at elevated salt levels). In a heterogeneous environment such as a lesion or the immediate vicinity of NETs, there may be proteases that can create such histone fragments. It may be a good idea for the authors to mention this just to cover the possibility.

We have addressed this possibility in a new discussion in lines 483-489.

2. Lines 124 – 128: The authors go through a chain of logic whereby they show that unlike LL37, antibiotics that act via protein synthesis inhibition but do not permeate membranes are not synergistic with H2A, which does not permeate membranes under low divalent conditions, therefore these observations indicate that the antimicrobial effect of H2A requires membrane permeabilization. At this point of the paper, perhaps the authors mean ‘consistent with’ and not ‘indicating’? strictly speaking, the conclusion is not possible until the localization studies of H2A done in the next section.

We changed the phrasing from “indicating” to “consistent with” (new lines 166-168).

3. The use of PROPS to monitor PMF is a nice touch. It will be helpful to the reader to point out potential limitations of this technique for the present purpose.

We have now included a more in-depth introduction of the PROPS method in lines 221-224 and included a brief discussion on potential limitations, in lines 228-230.

4. There are several points in the paper that can benefit from more clarifying text: For example, the PCA for the induced chromosome patterns, and a fuller discussion of the feedback loop model in Fig 5, so that the reader can better evaluate assumptions etc.

We have added a more detailed explanation of PCA in lines 338-346. We have added an explanation of the model in the main text in lines 399-403.

5. From the conclusions of the present paper, can we then say that an AMP like buforin, which is derived from H2A and can permeate membranes, self-amplifying?

The synergy effect is dependent upon pore formation and pore stabilization. An AMP which has both of these properties would be predicted to have a self-amplifying effect. H2A has the ability to stabilize pores but has relatively weak pore formation abilities and therefore is not self-amplifying. We note that at high concentrations, the pore formation effects of H2A are significant and the effects on bacterial death are consistent with self-amplification. We have added this discussion to the manuscript in lines 489-496.

Reviewer #2 (Remarks to the Author):

The manuscript by Doolin et al. focuses on the synergistic anti-microbial effects of histone 2A and the AMP LL37 in vitro. It provides a very interesting, novel concept for immune defence and covers different aspects of this mechanism. However, some important questions remain unanswered and some of the methods should be improved for what the authors are trying to show.

General concerns:

1. All experiments were conducted with H2A. Would the other histones be expected to have similar effects? This should at least be discussed.

The histones share structural similarities and chemical properties (DeLange et al., *Ann. Rev. Biochem.* 1971). Based on this, one would expect that other histones and histone-like fragments have bacterial killing activity. To more directly address the reviewer's question, we have performed new experiments with histone H3. As predicted by our previous results, synergy is observed between this histone and LL-37 (Fig. S1C). This suggests that synergy is general property between AMPs and histones. We now discuss these points in the discussion section in lines 478-496.

2. The authors mention NET formation as a mechanism where histones have an anti-microbial role. A hallmark of NET formation is global citrullination of histones. Would this alter the here-shown mechanism? It would be good to investigate this, especially if the authors discuss NET formation as a main source of histones and AMPs.

We thank the reviewer for raising this point. We note that the role of citrullination in NET formation is an area of active study, and has recently come into question in NETosis. Reports indicate that NET formation can proceed through a citrullination-independent process (Douda et al, PNAS, 2015 and Kenny et al., Elife, 2017). Regardless, the impact of citrullination on histone killing is an interesting question.

We have explored this by using citrullinated H3 (Fig. S1C). We observe a decrease in killing ability in citrullinated H3, but the synergy is still observed (discussed in lines 124-128). We note that while arginine residues are converted by citrullination, lysine residues are not, and that the content of lysine vs arginine residues differs between histones: H3 and H4 are much more arginine rich where as H2A and H2B have less. Thus, we would expect that citrullination would affect H3/H4's killing ability more than H2A's. This remains for future work to examine.

Given this likely difference between H3 and H2A, it seems possible that NETosis may involve a balance between DNA decondensation (driven by H3 citrullination) and retaining histone killing ability (affecting H2A/H2B less). This raises the possibility of different roles for histones in killing and NETosis, which we look forward to exploring in the future. We also note that histones also present in lipid droplets, which are ubiquitous, and there, are not subject to citrullination.

3. Also, other AMPs apart from LL37 should be considered. They are only very generally discussed.

We have conducted additional experiments with the pore-forming AMP magainin-2 and observed synergy between magainin-2 and H2A (Fig. 1H, 3C, 3F, S1I, S2B-C). Further, in support of our synergy model (where the pore-forming AMP increases entry of the histones), we note that we also observed synergy between H2A and the lipopeptide polymyxin B (Fig. 5C). Overall, the data support our proposed model of synergy between histones and AMPs. We have discussed the new results throughout the results section and discuss synergy between histones and other AMPs in lines 478-496.

4. How do the bacteria recover from LL37-H2A treatment? The growth curves indicate bacterial growth after approx. 15-18 hours -how is this possible? This should at least be discussed.

We attribute the resumption of growth to a small fraction of resistant mutants or phenotypic variants that are resistant or tolerant to the dual treatment of LL-37 and H2A. These mechanisms are responsible for antibiotic resistance (Lewis et al., Annu. Rev. Microbiol., 2010). We show that the resumption of growth is not unique to the LL-37/H2A treatment, but is typical following antibiotic treatment using chloramphenicol or kanamycin, which are routinely used in clinical

settings, at concentrations near or above the MIC (Fig. 1F, S1D). We have discussed this effect in lines 129-136.

5. The proposed mechanism of pore stabilization is not at all clear. Is there a direct interaction between the molecules? In my eyes, this is the main weakness of the paper.

We thank the reviewer for raising this point of clarification. We have clarified our definition of pore stabilization in lines 286-294, which was previously unclear. We define pore stabilization as the effect of H2A enhancing the permeabilizing effects of AMP-induced membrane pores by facilitating H2A and LL-37 uptake and by inhibiting the repair of the AMP-induced pores.

We do not yet know the exact mechanism of how histones stabilize pores. However, we have provided several lines of evidence of pore stabilization in previous and new experiments. We have shown that the presence of histones impedes the repair of pores formed by AMPs (Fig. 3E). We have performed a new experiment with a second AMP, magainin-2 and observed that H2A-treated cells don't recover from co-treatment with magainin-2 (Fig. 3C,F). In contrast, cells treated with only magainin-2 fully recover (Fig. 3C,F). We have performed an additional new experiment in low magnesium conditions, where histones can enter the cell (Fig. 3D) and found that cells can recover from the effects of H2A-induced pores. This indicates that the effect of pore stabilization is specific to pores formed by LL-37. Importantly, we have performed new experiments that determine the effect of internalized histones when membrane pores are absent by electroporating H2A into bacteria. We find that while histones have a detrimental effect on growth, cells can recover from the effects of H2A alone (Fig. 4A-B, S3A). In contrast, H2A and LL-37 in the same conditions leads to persistent permeation and the total inhibition of growth (Fig. 4A-B).

Finally, we have demonstrated that histone entry into cells inhibits global transcription (Fig. 4G-I). Taking this into account, the full data suggest that membrane permeabilization due to LL-37 persists due to the ability of histones to block the repair of membrane damaged caused by LL-37, but that this block involves some role for histones at the cell surface, in addition to any secondary role to block transcription.

Together, our evidence supports a model in which multiple mechanisms contribute to the pore stabilization effect by H2A that act at the membrane level or within the cytoplasm. At the membrane level, we propose two possible mechanisms, specifically that histones impede the repair of LL-37-induced pores through interactions at the pore, or that H2A increases membrane tension that keeps the pores open for longer times. In the cytoplasm, H2A inhibits membrane repair through the inhibition of transcription. We have discussed these models in the discussion in lines 453-477.

6. Chromatin rearrangement shown by Sytox Green staining is not convincing.

We note that this comment is repeated by the reviewer in point #20. We have addressed both comments here. We have performed new experiments that track HupA, which is a DNA-binding

protein that reports on the organization of the chromosome. We observed that H2A rearranges HupA-mRuby2 from a uniform distribution to a localization within the cell that is asymmetric with the long axis of the cell (Fig. 4E-F). This independently supports the idea that H2A rearranges the chromosome. We have discussed the new data in lines 347-355.

7. The importance of alterations of transcription (and translation?) at high concentrations of H2A remains unclear in relation to the proposed pore-formation action.

We note that the impact of H2A on transcription was performed not only at high (100 µg/mL) concentrations of H2A, but also for the physiological concentration used in most of the experiments in the manuscript (10 µg/mL). Our rationale for using the range of concentrations is that growth inhibition due to the specific effects of H2A is only achieved at high concentrations. At lower concentrations (10 µg/mL), H2A alone has no impact on growth (Fig. 1B,1D,4A,S1D,S3F,5C). The transcriptional response at lower concentrations of H2A alone thus do not necessarily provide a picture of how the cell responds to histones under stress. Importantly, our profiling across a range of H2A concentrations enables us to identify a trend in transcriptional response. The largest changes in transcription observed using 10 µg/mL H2A are also observed as the largest changes using 100 µg/mL H2A. In contrast, while H2A + LL-37 transcriptional data is relevant, it convolves the effects of H2A with potential AMP effects and limits the interpretation of the data. We have addressed the rationale behind this in lines 372-377 and lines 382-383.

8. In vivo toxicity of histones should be considered in the introduction and discussion.

We have added this to the introduction and discussion sections in lines 39-43 and in lines 497-502.

9. As mentioned above, other functions of histones in vivo and in the immune system should be discussed, such as the deleterious effects mediated through TLR2 and 4 (Allam et al., 2012, J Am Soc Nephrol., Xu et al., 2011, J Immunol.,) as well as their prominent role in sepsis (Xu et al., 2009, Nat Med). Also the role of histones in thrombosis should be considered (in the introduction or the discussion).

We have added the discussion of these issues in lines 39-43.

10. PI staining of bacteria incubated with H2A under low magnesium conditions: PI-positivity could also be the result of the bacteria undergoing apoptosis or of the membrane becoming more permeable as a secondary effect. How does this experiment show that H2A is actually inhibiting growth by primarily increasing cell membrane permeability, as the authors suggest?

We acknowledge that the increase in cell membrane permeability could be a secondary effect and that our data does not preclude this possibility. We have included a discussion of this when PI staining is first introduced in lines 97-100. We have performed new experiments that show that histones directly inhibit growth by electroporating histones into the cytoplasm (Fig. 4A,B,

and S3A). Further, RNAseq data from the H2A experiments indicate that bacteria upregulate membrane repair in response (Fig 4J and discussed now in lines 378-394), not apoptosis.

11. Figure 1C, figure legend: It would be helpful to indicate at what time point PI-fluorescence was measured in the legend. Generally, more information on treatment times in the figure legends would be good.

We have added this information in Figure 1C and all other relevant figure legends.

12. Figure 1A: *E. coli* appears to recover and proliferate after an initial inhibition, unlike *S. aureus*. How can this be explained? Is this difference attributable to the gram-status of the bacteria?

This effect could be due to *S. aureus*-specific physiology. Gram-positive bacteria may be more sensitive to histones in low magnesium environments. We have discussed this in lines 88-90.

13. In Figure 1E I find the way the bacteria are attached to each other after H2A treatment very intriguing, however I did not understand how this would happen. Is there any way to explain this very specific morphology? Especially in light of figure 2, where the authors show that H2A hardly enters the cell without formation of pores. How can this be explained?

We agree that this phenomenon is interesting. Previous works have noted that positively-charged molecules accumulate at the bacterial cell poles where the Gaussian curvature is highest (Huang, K.C. et al, Mol Micro 2010). The attachment here may be explained by large charge accumulation on the outside of the cells at the poles, which have now discussed in lines 151-157. We believe that in general, the aggregation of bacteria due to histones is interesting and will be the focus of future work.

14. In Figure 1E, it might be helpful to show lower power magnifications of the bacterial cultures as here it looks like there are more bacteria in the picture with H2A plus LL37 – I assume this is just a clustering effect, though?

We have added lower magnification images as Fig. S1F.

15. Also, please check Figure numbering – there is no Figure 1G, for example but it is mentioned in the text.

We have fixed the figure numbering.

16. Figure 1D: While there is an impressive delay in bacterial growth, after about 20 hours bacteria appear to be proliferating quite well again – do they develop some kind of resistance mechanism or how is the pore formation overcome?

We have fully addressed this in response to point #4 (reviewer #2).

17. Figure 2A: Fluorescence of bacteria treated with H2A alone should also be shown.

We have added this condition to the respective figure.

18. To claim that H2A enhances/facilitates pore formation by LL37 I don't think it is sufficient to measure cytoplasmic fluorescence of labelled LL37. Rather, it would be very helpful to actually show membrane localization of LL37, even if only in a semi-quantitative way, by higher resolution techniques.

In response to the reviewer's comments, we have added fluorescence images of bacteria with fluorescently-labeled LL-37 (Fig. 2E). We do not observe clear membrane localization of LL-37, which may be due to the internalized LL-37 signal swamping any membrane-bound signal. This observation is consistent with a previous report of LL-37 fluorescence, in which the membrane localization of LL-37 is transient (Snoussi et al, eLife, 2018). In addition, we observe that in treatments with LL-37 only, the fluorescence of the entire cell is significantly increased by the addition of H2A (Fig. 2E).

19. How would H2A stabilize LL37 pores? Would the two proteins interact directly, both of them being cationic? Has this been shown or can this be modelled? As the authors are stressing this synergistic action very strongly, more information on how this would occur needs to be provided. Would this interaction not change upon citrullination of histones in NET formation?

These points have been addressed in point #5 above (reviewer #2).

20. Figure 4B: The "reorganization of the chromatin" is not convincing. How do the authors know this is truly a reorganization and not just steric/static interference of H2A with the binding of Sytox Green?

We have fully addressed this in point #6 raised by reviewer #2 above through the use of the DNA-binding protein HupA to independently track chromosome organization.

21. Also, both the text in the Results section and the figure legend should mention for how long the cells were treated with H2A before the principal component analysis was performed.

We have added these details to both sections.

22. Finally, is there a way to perform a more functional analysis of the chromatin – as far as I understand, the principal component analysis mainly focuses on the staining pattern of the chromatin? Again, changes in this analysis might as well show altered staining behaviour instead of true reorganization of the chromatin.

We have fully addressed this in point #6 raised by reviewer #2 through the use of the DNA-binding protein HupA to independently track chromosome organization.

23. Figures 4D and E: How is mCherry fluorescence a specific indicator of transcription? A decrease in fluorescence could also indicate less translation, could it not?

Transcription of the mCherry reporter is controlled by the tetracycline promoter, which is activated by anhydrotetracycline. As such, this reporter measures the level of transcriptional activation. Similar constructs have been widely used to measure transcription in other studies. Nonetheless, we have added an acknowledgement that inhibition of translation could also affect mCherry fluorescence in lines 358-360. The inhibition of transcription by H2A is also supported by our measurements of total mRNA (Fig. 4H-I). Together, we believe that these data support the hypothesis that H2A inhibits transcription.

24. In Figure 4E, much higher concentrations of H2A than in the other experiments were needed to achieve an effect. Are these concentrations likely to be reached under physiological conditions? Do the authors think this is a relevant mechanism of action, if the other effects (pore stabilization, membrane depolarization) are seen at much lower concentrations?

This concern was partially addressed in our response to point #7 (reviewer #2). Higher concentrations were used to establish trends in transcriptional responses to H2A. We indeed observe that the same genes for membrane repair are upregulated at lower concentrations of histones (10 ug/mL). Histones are found in whole blood at 15 ug/mL following stimulation with *E. coli* in baboons (Xu et al., Nat. Med., 2009) and thus are a physiologically-relevant concentration. The concentration of histones that bacteria are exposed to when they are ensnared in NETs is unknown and could be potentially higher than the concentration in blood because NETs are a significant source of histones.

25. Why was RNA synthesis not investigated after treatment with LL37 plus H2A? As all other experiments were conducted in this fashion, this should be added.

We have added this data as Fig. 4H.

26. Discussion: The authors formulate a general defense mechanism for AMPs together with H2A. Other AMPs with similar effects should at least be discussed, if not shown in a proof of principle experiment.

This has been addressed in our response to point #3 (reviewer #2) in which we have performed additional experiments using the AMP, magainin-2. We point out that synergy with H2A was also observed in experiments with the pore forming lipopeptide polymyxin B (Fig. 5C). Finally, we have discussed the potential for other AMP synergies in lines 480-496.

27. As mentioned above, histones mediate significant toxicity in vivo, leading to organ damage, sepsis, thrombosis etc. This should be discussed here if application of histones is proposed as an antimicrobial therapy. Also, it should be considered that binding of histones and AMPs to NETs may be a mechanism to prevent generalized spread of histones (anti-inflammatory mechanism of aggregated NETs).

This has been fully addressed in the response to point #8 (reviewer #2).

Minor:

28. In the methods section, the authors sometimes use μl and sometimes ul .

We have made the corrections and made sure that μl is used in all instances.

29. There is no figure 1G, although it is mentioned in the text.

We have made this correction.

30. Line 257: delete the “a”.

We have made this correction.

Reviewer #3 (Remarks to the Author):

In this article, Doolin and collaborators explore the interactions of the histone H2A and the antimicrobial peptide LL-37. Co-treatment of *E. coli* with these two cationic proteins potentiates delayed bacterial growth, increased uptake of PI, and membrane blub formation. The results are expected since both antimicrobial proteins are well documented to act on the cytoplasmic membrane. The manuscript was hard to follow. Some of the supplementary figures were not commented, and main figures were out of order. Important experimental details were not included, making it difficult to interpret the data. Some of my concerns are:

1a. Fig. S1A is not commented in main text. Similar analyses should be presented as measure of antimicrobial activity of LL-32 and H2A. Figure 1E is wrongly referenced.

We have added a comment on Fig. S1A into the main text. We have performed an additional experiment to measure CFUs for LL-37 and H2A, as requested (now Fig. S1E). We have corrected the reference to Fig. 1E in the text.

1b. Some of the primary data are shown for the first time in Discussion.

We believe the reviewer is referring to the model. We have moved the model to the Results section.

2. Fig 1G is missing. SEM images don't clearly show membrane disruption. It is surprising that membrane blubs are not seen by SEM, when they seem to be formed in the bright fields.

We have fixed the reference to Fig. 1G. We have modified the description such that the SEM “suggest that the combination of LL-37 and H2A induces membrane damage” in lines 145-149.

We have also added additional SEM images that show structures that are consistent with membrane blebs (Fig. S1F). We note that membrane blebs are transient events (observed for only 10 minutes in Fig. 3A) such that they may not be observed in all cells. Together, the SEM

images, the PI fluorescence data, AF-H2A uptake, and fluorescent LL-37 uptake data together support the interpretation that the membrane is disrupted by the treatment of LL-37 and H2A.

3. Not clear how cell size was calculated.

We have added further details of cell size calculation in the methods section.

4. Increased antimicrobial activity of H2A in low magnesium is likely due to disruption of outer membrane and increased uptake of H2A into periplasm. This should be tested.

We have performed these experiments, added them as Figures 2C-D, and discussed the results in lines 199-201. Consistent with the reviewer predictions, low magnesium increases uptake of H2A into the cell.

5. The experiments with kanamycin (50 ug/ml!) are hard to interpret. At these high concentrations kanamycin causes death. The experiments should be done with doses closer to MIC. By the way, aminoglycosides cause membrane disruption through mistranslation of proteins. If the proposed model is correct, one would expect synergism between Km and H2A.

While the effects of kanamycin on membrane disruption are noted, it does not appear that kanamycin at the concentrations and incubation times used in this experiment causes sufficient membrane damage for PI to enter (Fig. 1E). In addition, it does not appear that the kanamycin-mediated membrane damage mechanism forms pores that are large enough for histones to enter the cell (Fig. 2B). Given the observation that kanamycin does not enable the entry of H2A into the cell, the model does not predict synergy between kanamycin and H2A. We have discussed this detail in lines 201-204. We have performed additional experiments using kanamycin at a lower concentration of 10 ug/mL, which is closer to the MIC (Fig. 1F). We do not observe synergy between H2A and kanamycin at this concentration.

6. Experiments in Figure 3 and 4 are missing important controls. What are the phenotypes of cells treated with LL-37 or H2A by themselves?

We have now added the controls to these figures for LL-37 and H2A by themselves (Fig. 3 and 4). We do not believe that the reviewer was requesting a LL-37 control for the RNAseq experiment (Fig. 4J), as this is beyond the scope of the manuscript.

7. The authors claim that H2A stabilizes pores formed by LL-37. However, the data do not conclusively show such a mode of action. It is equally possible independent pores by both antimicrobial peptides work together.

We have performed extensive additional experiments to address the issue of stabilization. We have discussed the new experiments (use of magainin-2, low magnesium, and electroporation) and clarified the model of stabilization in the response to point #5 for reviewer #2.

While our data suggest that histones form independent pores in conditions of low magnesium, the rest of the experiments of the paper in which synergy with AMPs is characterized,

physiological magnesium concentrations and 10 µg/mL of H2A are used. Here, our experiments show that H2A does not permeabilize membranes sufficient for PI or histones to enter the cytoplasm (Fig. 1E and 2B). Together with the finding that histones alone at 10 µg/mL do not affect growth, these data suggest that H2A does not form independent pores. We have discussed this in lines 209-210.

8. Most antimicrobial activity emanating from cotreatment of *E. coli* with LL37 and H2A appears to be manifested in the form of bacteriostasis, raising concerns about therapeutic potential as claim in the abstract.

We believe that our data are consistent with a bactericidal mechanism. We performed new experiments using plate-killing assays, which is typically used to determine whether agents are bacteriostatic or bactericidal. Bacteria were treated with LL-37 and H2A and plates on non-selective agar plates that did not contain histones or LL-37. We observed a significant decrease in colony-forming units when bacteria were treated with H2A and LL-37 (Fig. S1E). In addition, our recovery experiments following H2A and LL-37 treatment indicate that bacterial membranes do not recover from this treatment (Figs. 3B,E). Similar results were observed using H2A and magainin-2 (Figs. 3C,F).

We note that the increase in optical density at 15 hours following treatment with H2A and LL-37 (Fig. 1D) could be interpreted as bacteriostasis. However, cultures treated at near or above-MIC concentrations of kanamycin, which is a bactericidal drug, also increased at optical density at around the same time (Figs. 1F and S1D). Thus, the resumption of growth should not be interpreted as bacteriostasis but may be attributed to phenotypic variants that are resistant or tolerant (see response to point #4, reviewer #2).

Furthermore, bacteriostatic drugs (i.e., tetracycline, trimethoprim, erythromycin) are widely used in healthcare settings to date and can be effective as clinical treatments.

9. Figure 4A is unnecessary, since it is well known histones bind to DNA. The data could be moved to supplementary information.

We have moved this to the Supplement.

10. Line 201, Principal component analysis should be explained for people unfamiliar with the procedure.

We have addressed this in point #4 (reviewer #1) by adding a more detailed explanation of PCA.

Reviewers' comments:

Reviewer #1 (Remarks to the Author):

The authors answered my questions to my satisfaction.

The only point I want to posit is a comment in support of the authors on the relation between synergy/self-amplification and the role of NETs and citrullination, in questions raised by one of the other referees. Citrullination decreases charge on LL37 and histones and will necessarily negatively impact both their membrane remodeling ability and the DNA condensing ability of the latter. I don't think this is necessarily detrimental to the authors' argument: Components of NETs are widely implicated in autoimmunity. Having say PAD enzymes to citrullinate NETs components may be a natural way to modulate and control their activity. In any case, more work needs to be done in the field for this quite interesting and incompletely understood topic.

Reviewer #2 (Remarks to the Author):

The authors have made a significant effort to answer all previously raised concerns. The manuscript has certainly improved by the addition of many additional aspects and better controls, such as a second class of AMPs and by studying the effect of histone citrullination, just to give two examples. Some questions still remain (including the exact mechanism of pore stabilisation), however the discovery and characterisation of the synergistic effect of AMPs with histones is quite striking and, in my eyes, merits publication in Nature Communications.

The only aspect I still found somewhat unsatisfying was the novel LL37 staining. The authors here state that the cytoplasmic LL37 may have been swamping the membrane-bound LL37, however, as far as I can see, no confocal microscopy was performed? This would allow for a better visualisation of membrane-bound versus intracellular LL37. If the authors are worried about a transient localisation of LL37 to the membrane, perhaps a time-course could be performed? With and without histones? Also, higher resolution imaging of the membrane-bound LL37 should be evaluated (STORM/STED/EM with labelling?), although I understand this might not be feasible. I am sorry to emphasize this point so much but I think much of the here-presented chain of arguments strongly relies on this pore formation.

Other than that, I support publication in the journal and have no additional comments.

Reviewer #3 (Remarks to the Author):

The authors have done a very nice job at responding at my criticisms. The data provide a model for the apparent lack of antimicrobial activity of histones in culture at physiologic cation concentrations and show previously unsuspected, multifactorial interactions between histones and antimicrobial peptides. I still have a few minor concerns that should be considered before I endorse this paper for publication in Nat Communications.

Combined treatment of LL-37 and H2A considerably reduces CFU counts. These insightful data should be shown as part of figure 1 and not supplementary information.

Electroporation of H2A results in a marked lag phase. It is not clear if this is due to bacteriostasis or killing. Simple plating assays would not only discern between these two options but also inform about the relative weight membrane damage and DNA remodeling have on the antimicrobial activity of histones.

Line 165, it is true that some inhibitors of translation such as tetracycline or chloramphenicol do not affect membrane permeability. However, some classes of aminoglycosides do, as shown in references 61-63. The relatively low concentrations of kanamycin together with the short incubation times might have contributed to the negative results. Given these concerns I am still skeptical about the interpretation of this set of experiments.

I find the upregulation of colonic acid biosynthesis genes very interesting. The authors should consider that colonic acid helps Salmonella maintain the PMF (PMID 28588134). In other words, upregulation of colonic acid may represent an adaptation to drops in PMF.

We thank the reviewers for their interest in this work and for their helpful suggestions and comments. These comments and suggestions have been addressed as described below.

Reviewer comment are in **black**.
Author responses are in **blue**.

Reviewer #1 (Remarks to the Author):

The authors answered my questions to my satisfaction.

The only point I want to posit is a comment in support of the authors on the relation between synergy/self-amplification and the role of NETs and citrullination, in questions raised by one of the other referees. Citrullination decreases charge on LL37 and histones and will necessarily negatively impact both their membrane remodeling ability and the DNA condensing ability of the latter. I don't think this is necessarily detrimental to the authors' argument: Components of NETs are widely implicated in autoimmunity. Having say PAD enzymes to citrullinate NETs components may be a natural way to modulate and control their activity. In any case, more work needs to be done in the field for this quite interesting and incompletely understood topic.

We thank the reviewer for their insight and for pointing out how our results fit into the larger context of NET autoimmunity. We agree that citrullination may have an important role on NET antimicrobial activity and that moving forward, the field needs to determine the impact of citrullination-dependent and citrullination-independent NET formation on antimicrobial activity and autoimmunity. We have included statements to this effect in the discussion in lines 537-543.

Reviewer #2 (Remarks to the Author):

The authors have made a significant effort to answer all previously raised concerns. The manuscript has certainly improved by the addition of many additional aspects and better controls, such as a second class of AMPs and by studying the effect of histone citrullination, just to give two examples. Some questions still remain (including the exact mechanism of pore stabilisation), however the discovery and characterisation of the synergistic effect of AMPs with histones is quite striking and, in my eyes, merits publication in Nature Communications.

The only aspect I still found somewhat unsatisfying was the novel LL37 staining. The authors here state that the cytoplasmic LL37 may have been swamping the membrane-bound LL37, however, as far as I can see, no confocal microscopy was performed? This would allow for a better visualisation of membrane-bound versus intracellular LL37. If the authors are worried about a transient localisation of LL37 to the membrane, perhaps a time-course could be performed? With and without histones? Also, higher resolution imaging of the membrane-bound LL37 should be evaluated (STORM/STED/EM with labelling?), although I understand this might not be feasible. I am sorry to emphasize this point so much but I think much of the here-presented chain of arguments strongly relies on this pore formation.

We have performed new microscopy experiments to address the dynamics of LL-37 localization in the presence or absence of histones (Fig. 2G and S2D). Our current microscopy approach has

more than sufficient resolution to measure the localization of molecules to the bacterial membrane (see FM4-64 images in Fig. 2G and S2D). Our results show that LL-37 on its own does not localize to the membrane. In contrast, the co-treatment of LL-37 and H2A increases localization of LL-37 to the membrane over the course of 1 hour. These results provide further support that H2A enhances the effect of LL-37 on membranes. We have described these results in lines 228-237.

Other than that, I support publication in the journal and have no additional comments.

Reviewer #3 (Remarks to the Author):

The authors have done a very nice job at responding at my criticisms. The data provide a model for the apparent lack of antimicrobial activity of histones in culture at physiologic cation concentrations and show previously unsuspected, multifactorial interactions between histones and antimicrobial peptides. I still have a few minor concerns that should be considered before I endorse this paper for publication in Nat Communications.

Combined treatment of LL-37 and H2A considerably reduces CFU counts. These insightful data should be shown as part of figure 1 and not supplementary information.

We have moved this data (former Figure S1E in the Supplement) to Figure 1G in the main figures section.

Electroporation of H2A results in a marked lag phase. It is not clear if this is due to bacteriostasis or killing. Simple plating assays would not only discern between these two options but also inform about the relative weight membrane damage and DNA remodeling have on the antimicrobial activity of histones.

We have performed new CFU counting experiments and demonstrate that electroporation of H2A reduces bacterial growth by three orders of magnitude, to a limit that is nearly undetectable using standard CFU-counting methods (Fig. 4B). These experiments were performed on non-selective plates. Therefore, the results support that the mechanism of histone killing is bactericidal. Treatment using H2A in identical conditions but without electroporation shows a modest decrease in CFUs (Fig. 4B). This indicates that the entry of histones into bacterial cells is critical for antimicrobial activity. We have discussed these results in lines 325-331.

Line 165, it is true that some inhibitors of translation such as tetracycline or chloramphenicol do not affect membrane permeability. However, some classes of aminoglycosides do, as shown in references 61-63. The relatively low concentrations of kanamycin together with the short incubation times might have contributed to the negative results. Given these concerns I am still skeptical about the interpretation of this set of experiments.

We have modified our interpretation of the kanamycin experiments to acknowledge that the lack of synergy between H2A and kanamycin may be specific to our experimental conditions. We have added a statement that acknowledges that the incubation of H2A with higher concentrations

of kanamycin, longer incubation times, and different growth conditions could result in synergy. These are described in lines 175-182.

I find the upregulation of colonic acid biosynthesis genes very interesting. The authors should consider that colonic acid helps Salmonella maintain the PMF (PMID 28588134). In other words, upregulation of colonic acid may represent an adaptation to drops in PMF.

We thank the reviewer for their insight. We have added this to the discussion on lines 408-410.

REVIEWERS' COMMENTS:

Reviewer #2 (Remarks to the Author):

The authors have sufficiently answered all my questions. While I still do not agree that the resolution of images is very good, they do support what the authors want to show. Thus, I recommend publication in Nature Communications without further revisions.

Reviewer #3 (Remarks to the Author):

The authors have addressed my criticisms to my satisfaction

Reviewer comment are in **black**.
Author responses are in **blue**.

REVIEWERS' COMMENTS:

Reviewer #2 (Remarks to the Author):

The authors have sufficiently answered all my questions. While I still do not agree that the resolution of images is very good, they do support what the authors want to show. Thus, I recommend publication in Nature Communications without further revisions.

We thank the reviewer for their positive response. While not required, we have performed new experiments to increase the resolution of the LL-37 localization imaging by using a higher-resolution optical path on the microscope, increasing exposure time, and removing unbound fluorescent LL-37 from the medium. Our new images (now Fig. 3C) have significantly improved resolution and support our previous conclusions that H2A causes LL-37 to localize to the membrane.

Reviewer #3 (Remarks to the Author):

The authors have addressed my criticisms to my satisfaction

We thank the reviewer for their positive response.